# The phospholipid PI(3,4)P$_2$ is an apical identity determinant

Álvaro Román-Fernández [1,2], Julie Roignot[3,4,6], Emma Sandilands [1,2], Marisa Nacke[1,2], Mohammed A. Mansour [1,5], Lynn McGarry [2], Emma Shanks[2], Keith E. Mostov [3,4] & David M. Bryant [1,2]

Apical-basal polarization is essential for epithelial tissue formation, segregating cortical domains to perform distinct physiological functions. Cortical lipid asymmetry has emerged as a determinant of cell polarization. We report a network of phosphatidylinositol phosphate (PIP)-modifying enzymes, some of which are transcriptionally induced upon embedding epithelial cells in extracellular matrix, and that are essential for apical-basal polarization. Unexpectedly, we find that PI(3,4)P$_2$ localization and function is distinct from the basolateral determinant PI(3,4,5)P$_3$. PI(3,4)P$_2$ localizes to the apical surface, and Rab11a-positive apical recycling endosomes. PI(3,4)P$_2$ is produced by the 5-phosphatase SHIP1 and Class-II PI3-Kinases to recruit the endocytic regulatory protein SNX9 to basolateral domains that are being remodeled into apical surfaces. Perturbing PI(3,4)P$_2$ levels results in defective polarization through subcortical retention of apically destined vesicles at apical membrane initiation sites. We conclude that PI(3,4)P$_2$ is a determinant of apical membrane identity.

[1] Institute of Cancer Sciences, University of Glasgow, Glasgow G61 1BD, UK. [2] The CRUK Beatson Institute, Glasgow G61 1BD, UK. [3] Department of Anatomy, University of California, San Francisco, CA 94158-2140, USA. [4] Department of Biochemistry and Biophysics, University of California, San Francisco, CA 94158-2140, USA. [5] Biochemistry Division, Department of Chemistry, Faculty of Science, Tanta University, Tanta 31527, Egypt. [6] Present address: Broad Institute of MIT and Harvard, Cambridge, MA 02142, USA. These authors contributed equally: Álvaro Román-Fernández, Julie Roignot. Correspondence and requests for materials should be addressed to D.M.B. (email: david.bryant@glasgow.ac.uk)

The most common cell and tissue type is epithelium. The simplest epithelium is a monolayer of cells lining a biological cavity, such as a lumen. To generate such tissue, epithelial cells must form distinct cortical domains[1]. In a prototypical epithelium, the apical surface faces the lumen, the lateral surface interacts with neighboring cells, whereas the basal surface interacts with the extracellular matrix (ECM). The basal and lateral domains are contiguous and termed basolateral. The mechanisms controlling protein delivery to, and maintenance at, cortical domains in polarized cells have been extensively studied[2]. How epithelial cells become polarized and form a lumen de novo remains poorly understood, yet it is an outstanding problem in both development and disease.

MDCK cells grown inside ECM to form three-dimensional (3D) cysts have been widely used as a model system of polarization and lumen formation. In 3D, these undergo stereotyped morphogenesis, transitioning from a single cell to an apical-basal polarized monolayer radially organized around a central lumen[3]. During this process, each cell generates apical-basal polarization de novo. A number of polarization mechanisms first demonstrated in MDCK cysts are conserved in vivo[4–10]. Thus, MDCK cystogenesis is a powerful reductionist system to study epithelial polarization.

Upon 3D plating, single-MDCK cells divide into doublets with inverted polarity; some apical proteins, such as Podocalyxin/gp135 (Podxl), are found at the ECM-abutting surface but excluded from cell–cell contacts[11,12]. Integrin-dependent ECM sensing triggers Podxl endocytosis and transcytosis to the apical membrane initiation site (AMIS), a zone at doublet cell–cell contacts which remodels into the nascent lumen[13]. Remodeling involves conversion of a basolateral domain into an apical protein delivery zone. This stage is titled the pre-apical patch (PAP)[14]. The luminal space expands as the lumen matures. Delivery to the AMIS is regulated by the Rab11a GTPase. Rab11a influences molecular motors and vesicle docking and fusion machinery recruitment to ensure apical protein delivery to the AMIS[12,13,15–17]. Therefore, Rab11a-regulated exocytosis to the AMIS is crucial to generate apical polarity[1].

Phosphatidylinositol phosphate (PIP) asymmetry is essential for cell polarization[18]. PIPs can be modified by reversible phosphorylation of the 3-, 4-, or 5-position of their inositol ring[19]. Asymmetric PIP production at the cortex, or in organelles, determines membrane identity by scaffolding distinct PIP-binding proteins at these locales. In MDCK cysts apical-basal polarization depends on cortical PIP asymmetry regulated by the 3-phosphatase PTEN[11]: $PI(4,5)P_2$ is apically enriched, whereas $PIP_3$ is basolateral. This lead to a model proposing $PI(4,5)P_2$ as an apical identity determinant; this model is problematic, given that $PI(4,5)P_2$ is the precursor to $PIP_3$ and is also basolateral[11,18]. Whether alternate PIP species may fulfill an apical-specific function is unknown.

These advances focus attention on the key question of how existing cell surfaces are remodeled. Specifically, what controls cell–cell contact remodeling into an AMIS? We elucidate a molecular mechanism of de novo polarization through cortical PIP conversion to promote apical identity.

## Results

**PIP distribution during de novo apical-basal polarization.** De novo apical-basal polarization in MDCK cysts occurs via stereotyped stages (Fig. 1a)[11,12]. We examined PIP distribution during cystogenesis through fluorescent protein-fused PIP-binding domains[20]. In cysts with an open lumen, reporters for $PI(4,5)P_2$ were cortically distributed with apical enrichment, overlapping with apical Podxl (Fig. 1b, Supplementary Fig. 1a, white

arrowheads). In contrast, reporters for $PIP_3$ were basolateral (Fig. 1b, Supplementary Fig. 1a, white arrows), confirming previous results[11]. The $PI(4,5)P_2/PIP_3$ boundary was marked by Par3/aPKC (Fig. 1b, yellow arrowheads), the latter combination of which labels the AMIS during lumen initiation[12].

To determine when $PIP_3$ asymmetry emerges, we examined early lumenogenesis. At all stages, Par3 coincides with $PIP_3$ depletion. At the cell doublet stage, after initially peripheral Podxl internalized and transcytosed near cell–cell contacts, a single-Par3 punctum formed adjacent to each cell–cell contact (Fig. 1c, blue arrowheads), associated with $PIP_3$ depletion (white arrowheads). As the AMIS formed, the Par3-positive/$PIP_3$-depleted zone expanded. At the PAP, $PIP_3$ was further depleted and Par3 relocalized to the $PIP_3$/apical boundary (Fig. 1c, yellow arrowheads; Fig. 1b). This required the 3-Phosphatase PTEN, which localizes to the AMIS during polarity formation[21]. PTEN depletion attenuated lumen formation via subcortically stalling apically destined Podxl vesicles (Supplementary Fig. 1b–d). Thus, a PTEN-regulated Par3-positive/$PIP_3$-depleted zone marks the AMIS, the site for apical vesicle delivery.

We examined PIP content in transcytosing Podxl vesicles. While demonstrating different localizations, and some adjacency to Podxl/Rab11a vesicles, none of PI(3)P, PI(4)P, PI(5)P, $PI(4,5)P_2$, or $PIP_3$ showed strong localization to apically destined vesicles at the AMIS (Fig. 1d, yellow arrowheads), using multiple reporters where possible (Supplementary Fig. 1a, e-j). In contrast, whereas displaying some nuclear and cytoplasmic fluorescence, a probe for $PI(3,4)P_2$ (EGFP-2xPH-TAPP1[22]) overlapped specifically with Podxl staining at the apical surface, transcytosing vesicles, and the AMIS (Fig. 1e, white arrowheads). $PI(3,4)P_2$ may therefore be a regulator of apical domain function.

**PI(3,4)P₂ localizes to apical recycling endosome membranes.** We validated EGFP-2xPH-TAPP1 as a bona fide reporter of apical and recycling endosome $PI(3,4)P_2$. We developed a Pipeline for semi-automated phosphoinositide intensity analysis (PAPI), allowing robust identification of differential PIP localization across hundreds of cyst cultures (Fig. 2a).

EGFP-2xPH-TAPP1 expression, but not a phosphoinositide binding-deficient mutant (ΔPIP, TAPP1-2xPH-R211L[22]), was significantly apically enriched, and displayed nuclear recruitment (Fig. 2b, c), though the latter was not significant over nucleocytoplasmic protein GFP. In contrast, the PIP-mutant probe enriched basolaterally (Fig. 2c). Antibody staining for endogenous $PI(3,4)P_2$ displayed similar apical and nuclear localization (Fig. 2d, e), overlapping with the EGFP-2xPH-TAPP1 probe at the lumen (Fig. 2f), or periphery in cysts with inverted polarity (Fig. 2g), and was asymmetric to basolateral $PIP_3$ (Fig. 2h).

We characterized $PI(3,4)P_2$ localization during polarization using the EGFP-2xPH-TAPP1 probe which we will now refer to as $PI(3,4)P_2$. At all stages, $PI(3,4)P_2$ nuclear labeling is observed, but it will no longer be mentioned for ease of description. In early cyst development, $PI(3,4)P_2$ showed inverted localization, overlapping with peripheral Podxl and with Rab11a underneath apical membranes (Fig. 3a, b, white arrowheads). As Rab11a-positive vesicles rearranged towards the AMIS, these showed positivity for $PI(3,4)P_2$ (Fig. 3a, white arrows, Supplementary Fig. 2a, Supplementary Movie 1) and Podxl (Fig. 3b white arrowheads), displaying progressive $PI(3,4)P_2$ enrichment closer to the AMIS. As development proceeded from the PAP to open lumen, two PI(3,4)P₂ pools were apparent: Rab11a-positive endosomes (Fig. 3a, PAP, white arrows), and the lumen, the latter marked by F-actin/Podxl (Fig. 3a, b, white arrowhead; Supplementary Fig. 2b,

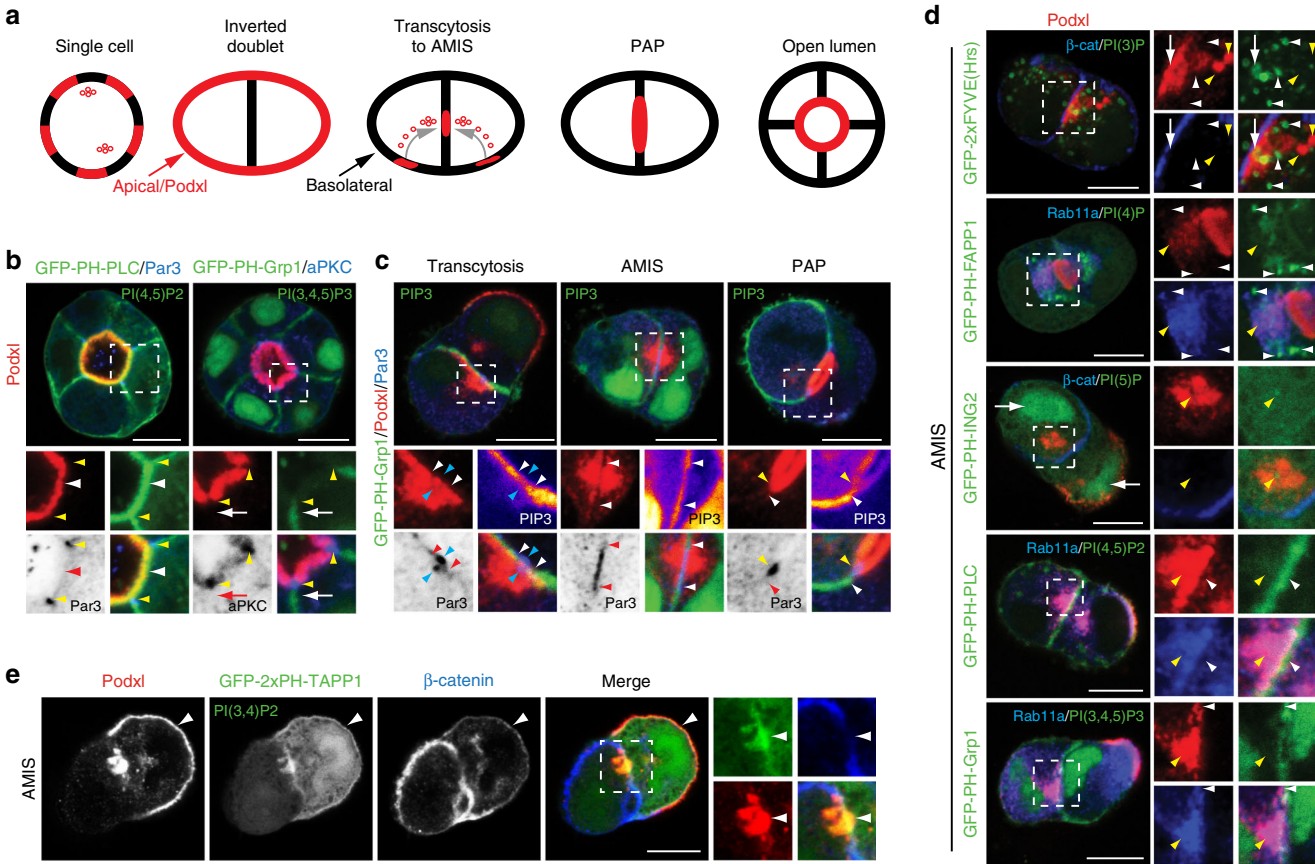

**Fig. 1** PIP distribution during polarization. **a** Cartoon of cyst development, showing progression from single cell through inverted doublet, transcytosing vesicles/AMIS (Apical Membrane Initiation Site), PAP (Pre-apical Patch) and Open Lumen stages. Red, apical/Podxl; black, basolateral. **b** PI(4,5)P$_2$ [EGFP-PH-PLC] and PI(3,4,5)P$_3$ [PIP$_3$, EGFP-PH-Grp1] (both green) localization in MDCK cysts with an open lumen (48 h), stained for Podxl (red) and either Par3 (left) or aPKC (right) (both blue or inverted greyscale). White arrowhead (red in inverted color panels), apical PI(4,5)P$_2$; white arrows (red in inverted color panels), basolateral PIP$_3$; yellow arrowheads, tight junction-localized Par3 or aPKC. In all instances, bottom panels are higher magnification of split color images from boxed regions. **c** PIP$_3$ [EGFP-PH-Grp1], Podxl and Par3 localization during lumen initiation (12–24 h). Note the single Par3 punctum forming adjacent the cell–cell contact in each cell (blue arrowheads). White arrowheads (red in inverted color panels), edge of PIP$_3$-enriched PM; yellow arrowheads, tight junction-localized Par3. PIP$_3$ in magnified boxes has been pseudocoloured to indicate intensity of labeling. **d** PI(3)P [EGFP-2xFYVE], PI(4)P [EGFP-PH-FAPP1], PI(5)P [EGFP-PH-ING2], PI(4,5)P$_2$ [EGFP-PH-PLC] and PIP$_3$ [EGFP-PH-Grp1] localization in cysts at the AMIS stage, stained for Podxl and either β-catenin (panels 1 and 3) or Rab11a (panels 2, 4, 5). Note the PI(3)P vesicles forming below the cell–cell contact near the β-catenin-depletion site (white arrow) and the absence of each PIP in Podxl-positive vesicles (yellow arrowheads). White arrowheads, PIP localization. **e** PI(3,4)P$_2$ [EGFP-2xPH-TAPP1] localization in MDCK cells at the AMIS stage, stained with Podxl and β-catenin. Arrowheads, Podxl at ECM-abutting surface and AMIS. Far right panels are higher magnification of split color images from the boxed region. All scale bars, 10 μm

Supplementary Movie 2). PI(3,4)P$_2$ is thus a component of apical membranes and recycling endosomes.

As we saw progressive Rab11a/PI(3,4)P$_2$ enrichment in vesicles towards the AMIS, we examined whether transcytosing Podxl transited via other PIP-enriched compartments. While PI(3)P lacked apparent overlap with Rab11a once lumens formed (Fig. 3c), PI(3)P-positive compartments (yellow arrowheads) closely apposed Podxl/Rab11a vesicles not yet clustered near the AMIS. In these locales occasional triple Rab11a/Podxl/PI(3)P positivity was observed (white/red arrows). In contrast, PI(4)P was observed both at the cortex (Supplementary Fig. 1f–h) and in Podxl-positive vesicular compartments (Fig. 3c, yellow/magenta arrows), which were separated from Rab11a-/Podxl-positive endosomes (white/red arrowheads). Instead, PI(4)P-positive structures were always adjacent to Rab11a. Podxl may pass through PI(3)P/PI(4)P-positive endosomes *en route* to the AMIS.

PI(3,4)P$_2$ has emerged as a regulator of clathrin-dependent and -independent endocytosis[23–25]. We observed that caveolin-1, but not clathrin heavy chain (CHC), co-localized extensively with PI(3,4)P$_2$ at the apical membrane at all stages (Fig. 3d, white arrowheads; Supplementary Fig. 2c, d), a site where the former overlaps with Cavin-1 (Supplementary Fig. 2e). Though intracellular caveolin-1 and PI(3,4)P$_2$ puncta occurred, these did not overlap (Fig. 3d, yellow arrowheads). These data reveal Rab11a and Caveolin-1, as markers of endosomal or apical pools of PI(3,4)P$_2$, respectively.

**PIP-modifying enzyme networks control polarization.** PIPs are made through combinatorial, reversible phosphorylation events[19] (Fig. 4a). We examined known PIP-modifying enzyme expression during polarization (Fig. 4b). 3D embedding of MDCK induces transcriptional reprogramming to express machinery required for 3D polarization[15]. We reasoned that this might extend to PIP-modifying enzymes. Comparison between monolayers (2D) and 3D cysts (23 h, 48 h) revealed extensive transcriptional changes to PIP-modifying networks, in a temporal fashion in 3D (Fig. 4b, Supplementary Fig. 3a). We performed functional

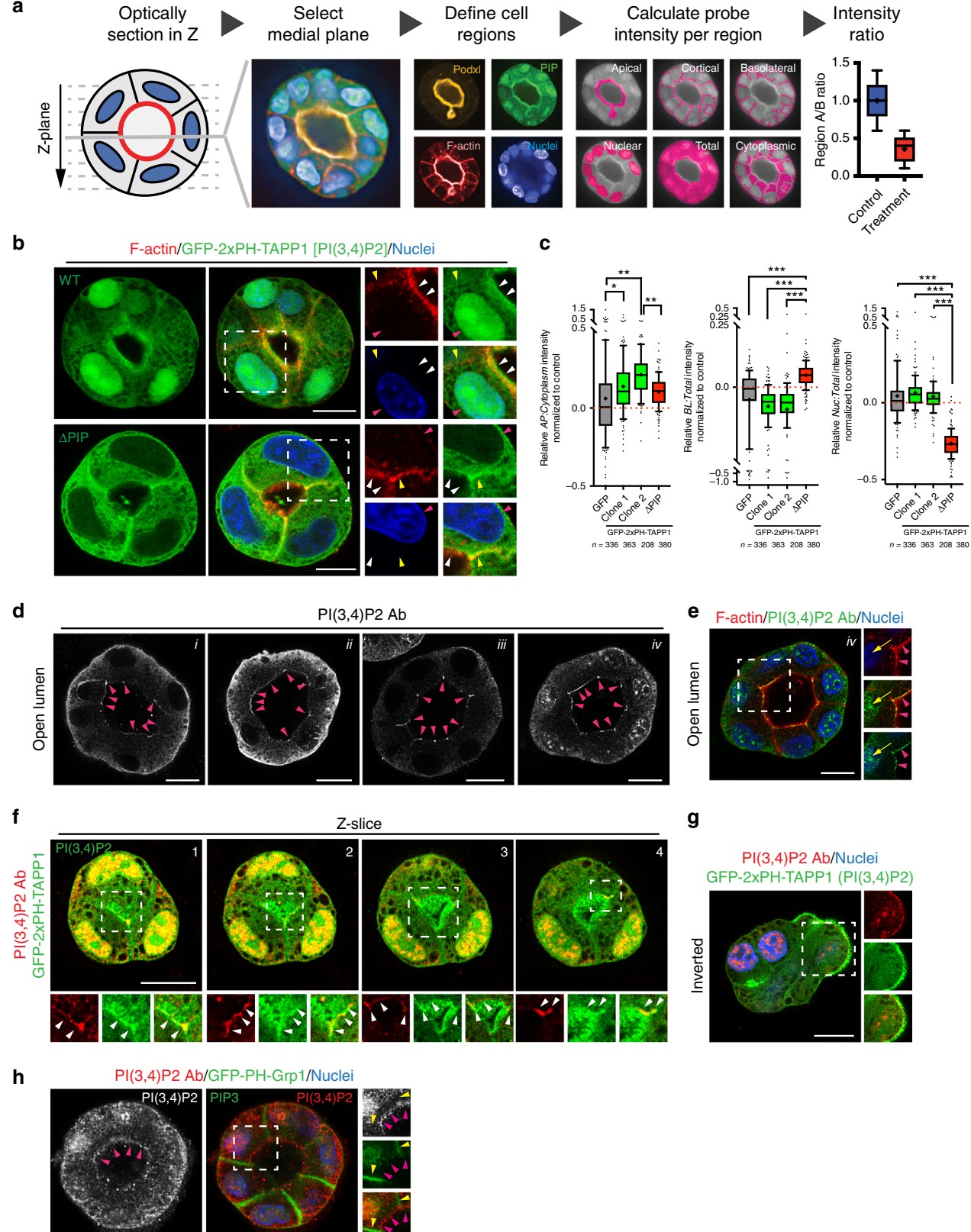

characterization of 5-phosphatases that might produce apical PI(3,4)P$_2$, INPP5E, SHIP1, and OCRL (Fig. 4b, red).

OCRL1-mCherry showed vesicular labeling with little overlap with Podxl at the AMIS or once lumens had formed (Supplementary Fig. 3b, yellow arrowheads). SGFP2-INPP5E

localized to a single punctum, occurring in the center of transcytosing Podxl vesicles, then juxtaposing the AMIS/PAP (Supplementary Fig. 3c, e, yellow and white arrowheads), before becoming localized to cilia and subapical puncta once lumens formed (Supplementary Fig. 3d). INPP5E depletion

**Fig. 2** Validation of PI(3,4)P2 localization. **a** Schematic representation of a Pipeline for semi-automated phosphoinositide intensity analysis, PAPI. MDCK cysts stably expressing GFP-tagged PIP reporters were cultured in 3D for 48–72 h, fixed and stained with Podxl to mark the apical domain, Phalloidin the cortex, and Hoescht the nucleus. Confocal optical sections of MDCK cysts were imaged and automated processing selected the medial plane based on the maximum lumen area. Separated cyst regions were defined based on differential localization of the above markers. PIP probe intensity was measured in each domain, followed by mathematical and statistical analysis to extract the relative PIP probe intensity ratio within compartments of the same object (cyst). **b** Cysts at the open lumen stage expressing either EGFP-2xPH-TAPP1 (WT) or a mutant EGFP-2xPH-TAPP1 unable to bind PI(3,4)$P_2$ (ΔPIP), stained for F-actin (red) and nuclei (blue). Luminal (white arrowheads), basolateral (yellow arrowheads), and nuclear (red arrowheads) localization is highlighted in magnified fields. **c** Quantitation of relative apical to cytoplasm (left), basolateral to total (center) or nuclear to total (right) PIP reporter intensity compared to GFP-overexpressing control MDCK cells. Box-and-whiskers: 10–90 percentile; +, mean; dots, outliers; midline, median; boundaries, quartiles. $n \geq 208$ cysts assessed from three wells/condition/experiment, three independent experiments (2 for EGFP-2xPH-TAPP1 WT clone 2). P-values (One-way ANOVA): *$P \leq 0.05$, **$P \leq 0.005$, ***$P \leq 0.0001$. **d**, **e** Forty-eight hours MDCK cysts stained for endogenous PI(3,4)$P_2$ (greyscale or green), F-actin (red) and nuclei (blue) using four different fixation and staining protocols, i[43], ii[12], iii[24], iv[44]. Note that PI(3,4)$P_2$ can be observed at the luminal domain in all cases (magenta arrowheads). Nuclear localization was also detected in some conditions (iv, **e**, yellow arrows). **f** Sequential optical sections of the medial region of an MDCK cyst expressing EGFP-2xPH-TAPP1 (green) stained for endogenous PI(3,4)$P_2$ (red). Note co-localization in the luminal membrane (white arrowheads) and nuclei. **g** Inverted polarized MDCK cyst expressing EGFP-2xPH-TAPP1 (green) stained for PI(3,4)$P_2$ (red) and Hoescht (blue). **h** Forty-eight hours MDCK cyst expressing a probe for PIP3 [EGFP-PH-Grp1, green] stained for endogenous PI(3,4)$P_2$ (red and greyscale) and Hoescht (blue). Yellow arrowheads, basolateral. Magenta arrowheads, luminal. All scale bars, 10 μm

attenuated lumen formation (Supplementary Fig. 3f–h), but not apical-basal polarization in filter-grown 2D monolayers (Supplementary Fig. 4a). As neither OCRL or INPP5E localize at the AMIS or show overlap with Podxl, it is unclear how it controls PIP3 > PI(3,4)$P_2$ conversion. We thus turned our attention to SHIP1.

**SHIP1 converts a cell–cell contact into an AMIS**. SHIP1 is a 5-phosphatase that converts PIP3 into PI(3,4)$P_2$[26], thereby triggering PI(3,4)$P_2$-dependent fast endophilin-mediated endocytosis[23]. SHIP1 showed robust, early upregulation in 3D (Fig. 4b). During polarization, SHIP1-EGFP co-localized with β-catenin-positive basolateral domains (Fig. 4c, blue arrowheads), becoming excluded from the forming lumen at the PAP stage (yellow arrowheads). Unexpectedly, after lumen formation, SHIP1-EGFP relocalized apically (yellow arrowheads). SHIP1 dynamic relocalization thus coincides with cortical depletion of its substrate, PIP3.

We inhibited SHIP1 by chemical and genetic means. To ensure robust polarity phenotype detection, we built a Pipeline for semi-automated polarity analysis, PAPA, quantifying hundreds to thousands of cysts per condition (Supplementary Fig. 4b), and performing comparably to manual quantitation (Supplementary Fig. 1c). SHIP1 chemical inhibition[27] at the time of plating strongly disrupted lumen formation, causing subcortical apical vesicle retention (Fig. 4d, Supplementary Fig. 4c). In contrast, chemical inhibition of SHIP1 after lumens formed failed to perturb cell polarity (Fig. 4d), in line with requirement to convert basolateral PIP3 to PI(3,4)$P_2$ only during AMIS formation. Endogenous SHIP1 depletion resulted in similar phenotypes (Fig. 4e–g). Control cysts displayed luminal Podxl (Fig. 4h, blue arrowheads), PIP3 reporter or β-catenin at basolateral membranes, and Par3 at the boundary between these zones (Fig. 4h, yellow arrowheads). SHIP1 depletion resulted in defective lumen formation and aberrant retention of PIP3, Par3, and β-catenin adjacent to rudimentary lumens (Fig. 4h, white arrowheads). SHIP1 PI(3,4)$P_2$-producing activity was required as only expression of RNAi-resistant WT, but not phosphatase-deficient, SHIP1 reversed SHIP1 depletion phenotypes (Fig. 4h, i). In contrast, filter-grown 2D monolayers displayed no defects in apical-basal polarization upon SHIP1 depletion (Supplementary Fig. 4a). SHIP1 is therefore a regulator of de novo apical membrane biogenesis in 3D.

We utilized PAPI to examine cortical PIP [PI(4)P, PI(3,4)$P_2$, PI(4,5)$P_2$, and PIP3] level and distribution after SHIP1 inhibition

(Fig. 5a–d). This confirmed exclusive localization of apical PI(3,4)$P_2$, cortical with apical enrichment PI(4)P and PI(4,5)$P_2$, and basolateral PIP3 (Fig. 5a–d). SHIP1 inhibition caused a significant, localized response, increasing basolateral PIP3 (Fig. 5b). We examined whether attenuating PIP3 generation (Class-I PI3-Kinase inhibition) could reverse the effect of PIP3 stabilization (SHIP1 inhibition). PI3-kinase inhibition (PI3K-i) has pleiotropic effects in 3D, as PIP3 regulates growth and polarity orientation[28–31], but also must be depleted at the AMIS. Class-I PI3K inhibition increases lumen formation in a population of cysts which display smaller size, and inversion of polarity in another pool (Fig. 5e–h). Consistent with a role in stabilizing PIP3 levels, SHIP1 inhibition (SHIP1-i) significantly increased cyst size, attenuated lumen formation and caused a mild, though significant, increase in polarity inversion. Combined SHIP1/PI3K inhibition failed to rescue polarity inversion, but completely rescued effects of SHIP1 on lumen formation and a significant, partial rescue of size (Fig. 5e–h). Regulation of basolateral PIP3 levels, and its metabolism into PI(3,4)$P_2$, are therefore essential to form a lumen de novo.

We examined whether exogenous PI(3,4)$P_2$ supply would promote lumen formation (Fig. 5i). Strikingly, addition of exogenous PI(3,4)$P_2$ to cysts significantly increased lumen formation without affecting polarity inversion, but only if SHIP1 was not inhibited. Thus, while AMIS-localized PI(3,4)$P_2$ generation promotes lumen formation, co-depletion of PIP3 is essential.

**Class-II PI3-kinases contribute to apical polarization**. Following PIP3 to PI(3,4)$P_2$ conversion at the AMIS, SHIP1 was dispensable for polarity (Fig. 4d). We examined the contribution of the endosomal PI(3,4)$P_2$ pool to lumen formation. PI(3,4)$P_2$ can also be generated from PI(4)P by Class-II PI3-kinases[24,32]. We noted upregulation of PIK3C2G in 3D, and downregulation of PIK3C2A/B (Fig. 4b). Unexpectedly, PIK3C2A and PIK3C2B depletion perturbed lumen formation, whereas depletion of PIK3C2G modestly promoted lumen formation (Fig. 6a–c). PIK3C2A/B co-depletion robustly inhibited lumen formation (Fig. 6a–c). Thus, PIK3C2A/B are non-redundant lumen formation regulators.

PIK3C2A/B displayed distinct localizations, though both overlapped with different pools of Rab11a. GFP-PIK3C2A was closely associated with Rab11a vesicles throughout polarization (Fig. 6d). A pool of GFP-PIK3C2B was basolateral and

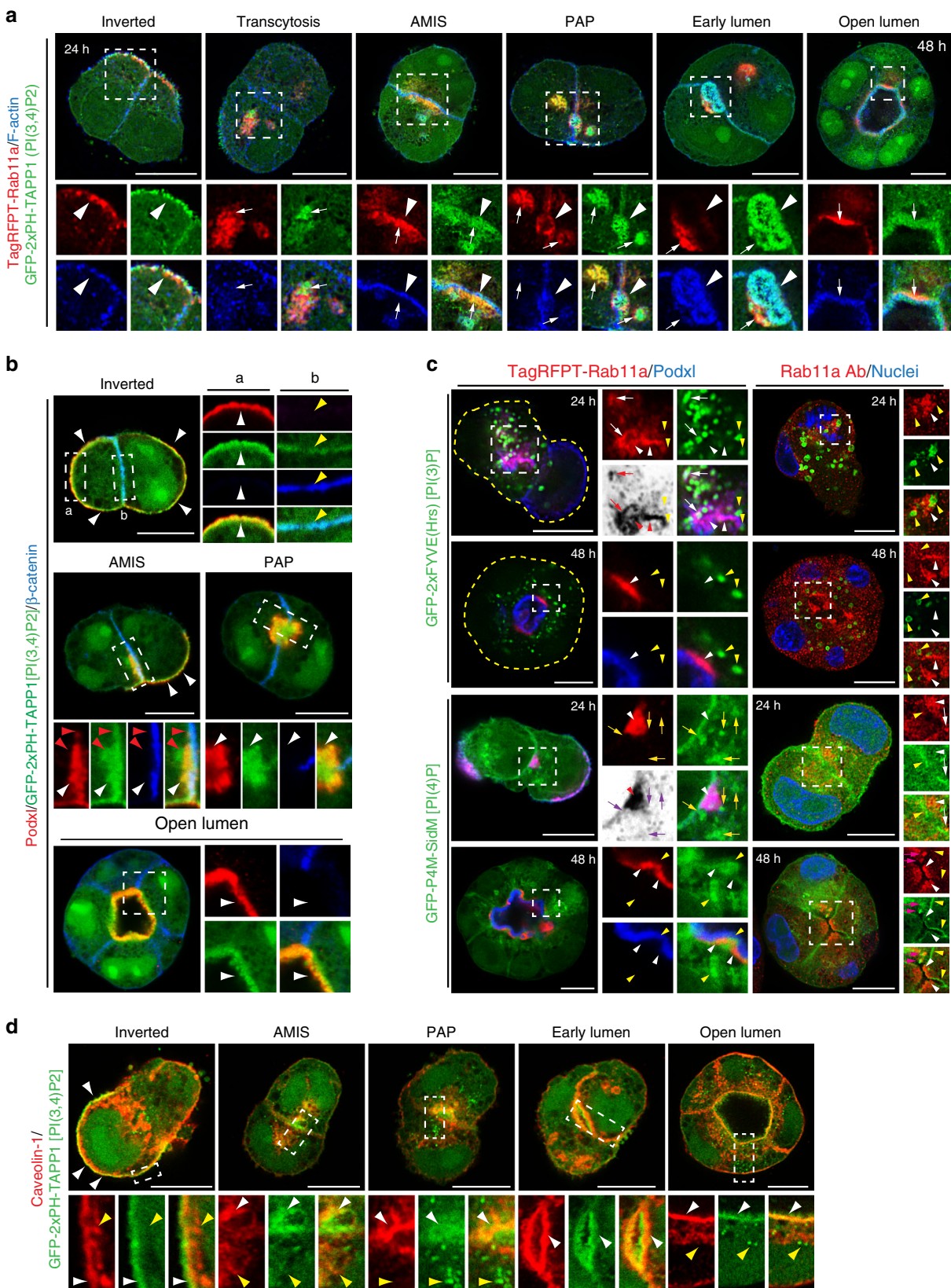

overlapped with clusters of Rab11a that were close to the cortex (Fig. 6d, e). GFP-PIK3C2B was initially at doublet cell–cell contacts, but became progressively removed from the forming AMIS (Fig. 6f). In cysts with an open lumen, GFP-PIK3C2B could be seen both at the basolateral membrane and in vesicular compartments near Rab11a. Thus, PIK3C2A is constitutively Rab11a-endosome adjacent, while PIK3C2B may be involved in cortical Rab11a events controlling apical polarization.

We examined whether PIK3C2A/B could counteract SHIP1 inhibition. Despite both kinases being required for lumen

**Fig. 3** PI(3,4)P$_2$ is an apical- and recycling endosome-enriched lipid. **a**, **b** PI(3,4)P$_2$ localization during different stages of lumen formation (12–48 h) in MDCK cells stably co-expressing EGFP-2xPH-TAPP1 [PI(3,4)P$_2$ sensor, green] and either **a** TagRFPT-Rab11a WT (red) and stained for F-actin (blue), or **b** Podxl (red) and stained for β-catenin (blue). White arrows, PI(3,4)P$_2$ co-localization with Rab11a vesicles at all stages; white arrowheads, cortical PI(3,4)P$_2$ localization; red arrowheads, PI(3,4)P$_2$-positive, Podxl-negative vesicles below the cell–cell contact at the AMIS stage; yellow arrowheads, PI(3,4)P$_2$ present at low levels at the basolateral PM at the AMIS stage. In all instances, bottom panels are higher magnifications of split color images from the boxed regions. **c** Localization of Rab11a (TagRFPT-Rab11a, left; endogenous antibody staining, right; both red) in either 24 or 48 h cysts in relation to either PI(3)P [EGFP-2xFYVE, top] or PI(4)P [EGFP-P4M-SidM, bottom] (both green). In blue, Podxl (left panel) or nuclear staining (right panel). Yellow arrowheads, Rab11a-negative PIP reporter vesicles. White arrowheads (red in inverted color panels), Rab11a endosomes. White arrows (red in inverted color panels), triple Rab11a/Podxl/PIP reporter-positive vesicles. Yellow arrows (magenta in inverted color panels), Rab11a-negative, Podxl/PIP reporter-positive vesicles. Pink arrows, Rab11a/PI(4)P overlapping endosomes. **d** PI(3,4)P$_2$ and caveolin-1 localization during lumen formation (12–48 h) in EGFP-2xPH-TAPP1 (green) MDCK cells stained with Caveolin-1 (red). Note the cortical co-localization of Caveolin-1 and PI(3,4)P$_2$ at all stages (white arrowheads) and the absence of intracellular co-localization (yellow arrowheads). All scale bars, 10 μm

formation (Fig. 6a–c), only GFP-PIK3C2A overexpression increased lumen formation, similar to Rab11a overexpression[12], and rescued SHIP1 inhibition (Fig. 6g). Thus, PIK3C2A/B may promote lumen formation from different pools, with the Rab11a/PIK3C2A pool increasing de novo apical domain generation.

**INPP4A/B are negative regulators of apical polarization.** As lumen formation efficiency could be both increased and decreased, we looked for potential tonic PI(3,4)P$_2$ regulators. The 4-phosphatases INPP4A/B, which can convert PI(3,4)P$_2$ to PI(3)P[33], were initially downregulated in 3D, with INPP4A re-induced at 48 h (Fig. 4b). Ectopic expression of INPP4B significantly attenuated single lumen formation, causing subcortical Podxl vesicle accumulation, in a phosphatase-dependent fashion (Fig. 7a–c). Conversely, depletion of INPP4A/B alone or together increased lumen formation (Fig. 7d–f), mirroring PIK3C2A overexpression (Fig. 6g). Unlike PIK3C2A overexpression however, INPP4A/B depletion alone or together failed to rescue the effect of SHIP1 inhibition (Fig. 7f). Thus, transcriptional downregulation of PI(3,4)P$_2$-depleting enzymes is essential for de novo lumenogenesis.

**The apical Par complex is PI(3,4)P$_2$-independent.** We aimed to identify the functional consequence of apical PI(3,4)P$_2$ generation. The PTEN-mediated production of apical PI(4,5)P$_2$ from PIP$_3$ controls the function of the aPKC/Par6/Cdc42 module in association with apical Annexin2[11]. In contrast, SHIP1-mediated PI(3,4)P$_2$ production did not affect aPKC level or activation (Fig. 8a), Annexin2 apical, cortical or apical-basal localization ratio (Fig. 8b, c), recruitment of Cdc42 onto Podxl/Rab11a-positive vesicles (Fig. 8d), activation levels of Cdc42 at the apical membrane (Fig. 8e) or recruitment of WT or mutationally active Cdc42 to the cortex, to the apical domain or the apical-basal ratio (Fig. 8f). Thus, production of PI(3,4)P$_2$ from PIP$_3$ at the AMIS by SHIP1 does not regulate the apical aPKC/Cdc42/Par3 complex.

**PI(3,4)P$_2$-SNX9 interaction is essential for lumen formation.** To identify PI(3,4)P$_2$-interacting protein(s) regulating polarization, we examined whether INPP4B overexpression disrupted known PI(3,4)P$_2$ interactor localization[24]. We focused on labeling endogenous proteins, to preclude counteraction of INPP4B overexpression by exogenous effector protein co-overexpression. We detected that endogenous SNX9, a protein involved in PI(3,4)P$_2$-dependent endocytosis, was present in subapical puncta that were lost upon INPP4B overexpression (Fig. 9a). Depletion of SNX9 phenocopied SHIP1 depletion or INPP4B overexpression, resulting in subcortical Podxl accumulation and defective lumen formation (Fig. 9a–c). Unlike WT SNX9 expression, PI(3,4)P$_2$ binding-deficient SNX9 mutants[24] failed to reverse the effect of SNX9 depletion (Fig. 9d–f). Combining SNX9 depletion with

PIP-binding-deficient SNX9 expression caused subcortical vesicular accumulation of non-overlapping pools of Podxl and β-catenin, but both of which did overlap with mutant SNX9. Therefore, the formation of PI(3,4)P$_2$ at the AMIS is essential for SNX9 to remodel basolateral components to form an apical domain de novo.

**Discussion**

We describe a transcriptionally regulated PIP-modifying enzyme network essential for lumen formation. Notably, the enzymes required in 2D and 3D are not interchangeable, though both conditions are apical-basal polarized. This revealed an unanticipated role for the PI(3,4)P$_2$ as a determinant of 3D apical identity. Specific to the 3D context is basolateral cell–cell contact remodeling into an apical domain[12]. Several PIP-modifying enzymes participate in this process.

The doublet cell–cell contact contains a PI→PI(4)P→PI(4,5)P$_2$→PIP$_3$ cascade that generates basolateral identity (for model, see Fig. 10a–c). During polarity rearrangement, PTEN locally reverses the last step in the PIP$_3$ cascade, inducing a PIP$_3$ depletion zone: the AMIS[11,21] (Fig. 10b). Due to this early apical enrichment, PI(4,5)P$_2$ was described as an apical targeting factor[34]. However, PI(4,5)P$_2$ is also basolateral. Our current data suggest an updated model wherein lack of PIP$_3$, rather than the mere presence of PI(4,5)P$_2$, encodes part of apical identity.

We report that the long-unidentified apical domain identity determinant is PI(3,4)P$_2$, which is enriched apically and on recycling endosomes. That PI(3,4)P$_2$ is asymmetric in localization and function to PIP$_3$ is unexpected. PI(3,4)P$_2$ was considered as part of PIP$_3$ signaling cascades, given that both lipids bind the Akt PH domain in vitro[35]. Such studies have examined PI(3,4)P$_2$ in single or poorly polarized cells, where apical domains may not occur. The functions and asymmetry of PI(3,4)P$_2$ may have been underappreciated in such contexts.

How is PI(3,4)P$_2$ generated? At least two pools of PI(3,4)P$_2$ exist and these dynamically rearrange during polarization: Rab11a endosomes and apical surfaces. In early cell doublets (Fig. 10a), PI(3,4)P$_2$ is enriched at the periphery with apical proteins. As polarity reorientation is triggered, Podxl-positive vesicles transcytosed to the AMIS. These vesicles initially contain PI(3)P/PI(4)P, before enriching with PI(3,4)P$_2$ and Rab11a as Podxl clusters at the AMIS. The class-II PI3-kinases PIK3C2A/B, which generate PI(3,4)P$_2$ from PI(4)P in other contexts[24,32], may promote formation of endosomal PI(3,4)P$_2$, though acting in different locales: PIK3C2B localizes with Rab11a puncta close to the cortex, whereas PIK3C2A localizes closely opposed to transcytosing Rab11a. Rab11a and PIK3C2A likely co-operate to promote PI(3,4)P$_2$ as overexpression of either, or exogenous supply of PI(3,4)P$_2$, increases lumen formation rates. Indeed, PIK3C2A controls Rab11a activation in other systems[36]. It is

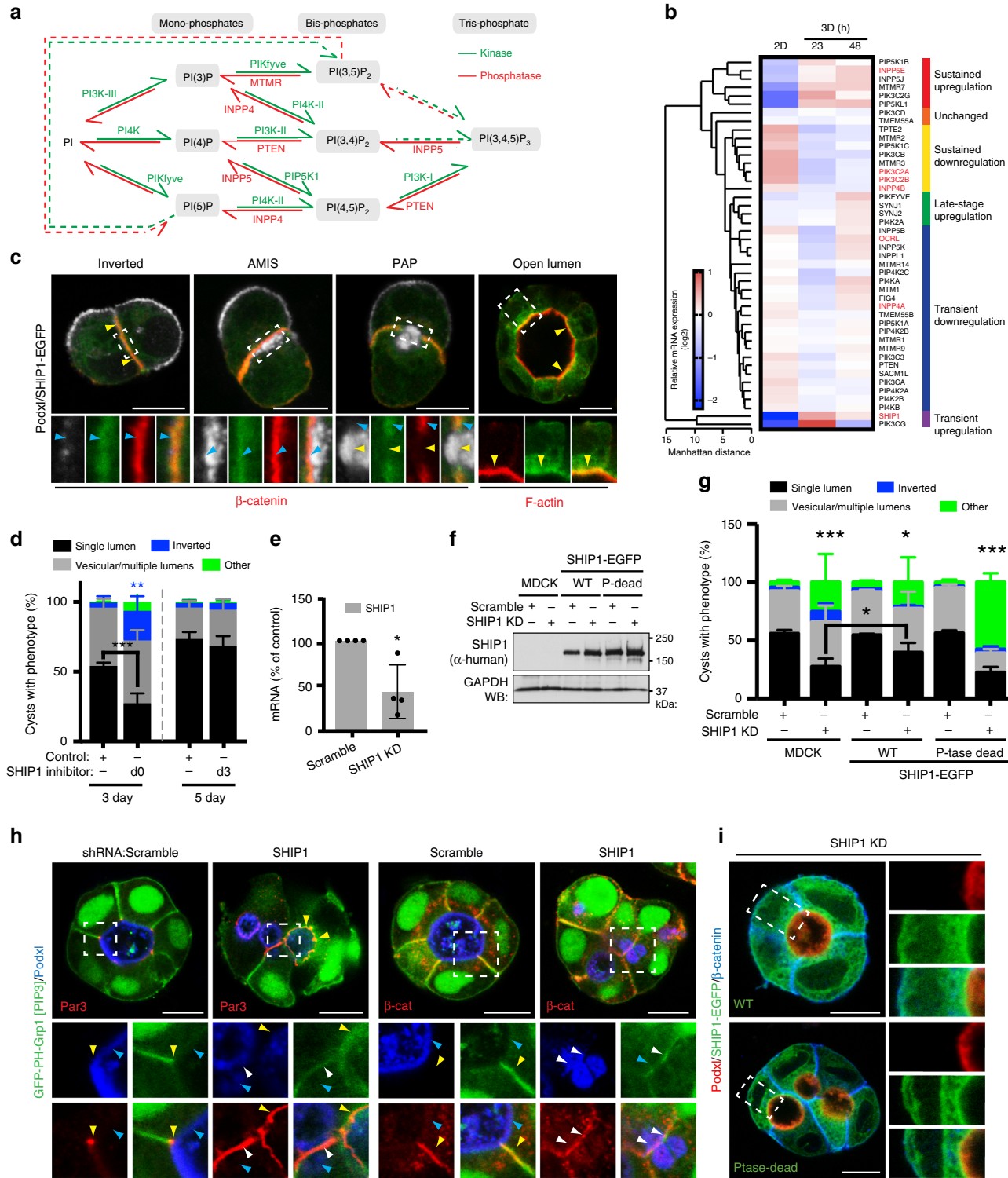

important to note that INPP4A/B, which can decrease PI(3,4)P$_2$ through conversion to PI(3)P[35], were transcriptionally down-regulated in 3D during the developmental window of apical vesicle transcytosis. Ectopic re-expression of INPP4B disrupts apical polarization in a phosphatase-dependent fashion. Concomitant with transcytosing apically destined vesicles, transcriptionally upregulated SHIP1 is responsible for the initial conversion of basolateral PIP$_3$ at the AMIS into PI(3,4)P$_2$. Once this event occurs, SHIP1 is now dispensable for polarity, and the

apical enrichment of PI(3,4)P$_2$ is likely reinforced by delivery of PI(3,4)P$_2$-positive Rab11a vesicles.

PTEN can also function as a PI(3,4)P$_2$ phosphatase[37]. Given that PTEN is enriched at the forming AMIS[21], whether it is acting solely on PIP$_3$ or also on PI(3,4)P$_2$ is unknown. If both are occurring, the AMIS localization of PIK3C2B may help to focally reverse this dual specificity, leading to co-enrichment of apical PI(4,5)P$_2$ and PI(3,4)P$_2$. The exact function of PTEN at the AMIS thus warrants further attention.

**Fig. 4** SHIP1 converts PIP$_3$-rich basolateral membrane into apical domains. **a** Conversion between the phosphoinositide species, all derived from phosphatidylinositol (PI), occur via the action of kinases (in green) and phosphatases (in red). Dashed arrows, pathway whose occurrence or regulatory enzyme is still unknown. **b** Heat map of Manhattan-clustered, differentially expressed PIP kinases and phosphatases in cells grown as a monolayer for 48 h or as cysts in Matrigel for 23 or 48 h. Relative mRNA expression levels (log2 values), and clustering categories, are shaded as indicated. Four independent experiments. **c** SHIP1 localization during lumen formation in MDCK cells expressing SHIP1-EGFP (green) and stained for Podxl (white) and β-catenin or F-actin (red). **d** Quantitation of cyst phenotypes treated with ethanol control or SHIP1 inhibitor either at the time of plating (d0) and fixed at day 3 (3 day, left), or treated at day 3 (d3) and fixed at day 5 (5 day, right). Values are mean ± s.d. For 3 day, $n \geq 300$ cysts assessed from three wells/condition/ experiment, four independent experiments, P-values (two-way ANOVA): **$P \leq 0.001$, ***$P \leq 0.0001$. For 5 day, $n \geq 1900$ cysts assessed from three wells/ condition/experiment, three independent experiments. **e** RNA extracts from MDCK cells stably expressing scramble or SHIP1 shRNA were analyzed by RT-qPCR to detect *SHIP1* mRNA levels ($n = 1$ well per condition, from four independent experiments). P-value (Student's t-test): *$P \leq 0.05$. **f, g** Western blot of WT and phosphatase-dead SHIP1 and GAPDH in total cell lysates of parental (MDCK) or SHIP1-EGFP-expressing cells expressing scramble or SHIP1 shRNA **f**, and quantitation of cyst phenotypes **g**. Mean ± s.d., $n \geq 400$ cysts from four wells/condition/experiment. P-values: two-way ANOVA): *$P \leq 0.05$, **$P \leq 0.001$, ***$P \leq 0.0001$. **h** PIP$_3$ [EGFP-PH-Grp1], Podxl and Par3 (left panels) or β-catenin (right panels) localization in cysts expressing scramble or SHIP1 shRNA. Yellow arrowheads, Par3 (left panels) or β-catenin (right panels); blue arrowheads, Podxl. White arrowheads, overlap of apical and basolateral domains. **i** Immunolabelling of above conditions, with Podxl and β-catenin. Two independent experiments for parental MDCK versus SHIP1-EGFP WT cells and one experiment for parental MDCK cells versus SHIP1-EGFP WT cells or SHIP1-EGFP phosphatase-dead cells. Scale bars, 10 μm

What is the function of apical PI(3,4)P$_2$ in 3D polarization? In contrast to PTEN-mediated PIP$_3$ to PI(4,5)P$_2$ conversion, SHIP1-mediated PIP$_3$ to PI(3,4)P$_2$ conversion did not affect apical aPKC-Cdc42 activity. In cells in 2D, PI(3,4)P$_2$ regulates cell motility, and both clathrin-dependent and -independent endocytosis[23–25,38,39]. In our system, apical PI(3,4)P$_2$ most closely overlapped with Caveolin-1, and not clathrin, suggesting a clathrin-independent pathway may remodel basolateral cargo proteins through endocytosis to allow the AMIS to form. Accordingly, perturbing PI(3,4)P$_2$ levels or its interaction with SNX9 results in the subcortical vesicular localization of both apical and basolateral cargoes. One function of cortical PI(3,4)P$_2$, in addition to promoting a zone for apical exocytosis, may be to promote endocytosis of basolateral proteins at the AMIS. The function of recycling endosome-localized PI(3,4)P$_2$ is unknown, but may be to recruit other sorting nexins, such as the apical exocytosis-promoting SNX9 homologue, SNX18[17].

If PI(3,4)P$_2$ is both in recycling endosomes and the apical domain, how can it be an apical determinant? It may be that discrimination between these two PI(3,4)P$_2$-positive compartments is provided by the absence or presence of PI(4,5)P$_2$, respectively. Such a combinatorial PIP code for membrane identity greatly expands the possible number of compartment identities than can be generated by the seven PIPs alone. In line with this, although exogenous addition of PI(3,4)P$_2$ enhanced lumen formation efficiency, it could only do so when SHIP1 was functional. The apical identity code may therefore be the presence of PI(3,4)P$_2$ and PI(4,5)P$_2$, and the absence of PIP$_3$. In support of this, exogenous PIP$_3$ addition to an apical membrane causes rapid conversion to basolateral identity[28]. The recycling endosome code may rather be PI(3,4)P$_2$ without PI(4,5)P$_2$.

Our studies underpin that whereas there are in theory multiple ways to make a PIP, these are transcriptionally and spatio-temporally regulated during development. What is important to note is that PIP production and morphogenesis go hand-in-hand: inhibiting SHIP1-mediated PIP$_3$ to PI(3,4)P$_2$ conversion results in basolateral domains with more PIP$_3$, but not with apical domains with less PI(3,4)P$_2$. Rather, an apical domain will not form if sufficient apical PI(3,4)P$_2$ is not generated, or if cortical PIP$_3$ is not removed. This represents a conceptual difference from 2D studies where the cortex is a constant structure where PIP levels can change. In multicellular 3D contexts, such changes in cortical PIP levels induces alternate morphogenesis, such as conferring apical or basolateral identity.

In addition to their function in cell polarization, phosphoinositides participate in cell growth and survival pathways[18,40]. For instance, correct cell polarization may lay upstream to decisions of cell death or survival during morphogenesis. Likewise, perturbation of PIP-modifying enzymes, such as the PTEN or INPP4B loss observed in cancer[40,41], may be facilitative of the disrupted polarity and overgrowth of tumors. The extent to which these processes are distinct pathways or are intimately linked remains unclear.

We describe an unappreciated function of PI(3,4)P$_2$ in apical domain morphogenesis in MDCK cysts. Although an in vitro reductionist system, the molecular mechanisms elucidated in MDCK cysts have been demonstrated as conserved in a variety of other systems[4–9]. Yet, there are several developmental mechanisms for lumen formation; the polarized exocytosis and membrane remodeling we describe here can be bypassed by apoptotic cavitation of multicellular clusters or folding and joining of tissue sheets[1,42]. In these instances, we predict that apical PI(3,4)P$_2$ is likely supplied by the endosomal pool, rather than conversion from basolateral PIP$_3$ by SHIP1. In spite of differences in the way it can form, the apical domain is essential for exchange of proteins, nutrients, solutes and lipids. Given, for example, that the kidney lumen is an extremely active site of endocytosis in vivo, that PI(3,4)P$_2$ has emerged as a regulator of rapid endocytosis aligns with such a physiological requirement[23–25]. It will thus now be important to determine the involvement and regulation of apical and recycling endosome PI(3,4)P$_2$ in development and disease.

## Methods

**Cell and cyst culture**. MDCK-II (K.Mostov, UCSF) cells were grown in 5% fetal bovine serum (FBS; Gibco) in minimum essential medium (MEM, Gibco). 293-FT cells (Thermo Fisher Scientific) were cultured in Dulbecco's modified Eagle's medium (DMEM, Gibco) supplemented to final 10% FBS, 0.1 mM non-essential amino acids, 1 mM sodium piruvate and 6 mM L-glutamine. 293 GPG cells were grown in DMEM supplemented to final 10% heat-inactivated FBS, 2 mM L-Glutamine, 20 mM HEPES, 1 μg ml$^{-1}$ tetracycline (Sigma T-7660) and 2 μg ml$^{-1}$ puromycin (Sigma P-7255) (K.Mostov, UCSF) Culture of MDCK-II as 3D cysts was adapted from previous protocols[13]. Briefly, single cell suspensions of $1.5 \times 10^4$ cells ml$^{-1}$ in medium containing 2% Matrigel (BD Biosciences) were plated either onto 8-well coverglass chambers (Nunc, LabTek-II) or 96-well black bottom plates (Greiner, SLS); 300 or 150 μl total, respectively), pre-coated with 5 μl of Matrigel (100%). For SHIP1 inhibition experiments, cells were treated from the time of plating, unless otherwise specified, with either 5 μM SHIP1 inhibitor (3AC, Calbiochem) or ethanol as a control. For Type I-PI3K inhibition experiments, cells were treated from the time of plating with 10 μM of type I-kinases inhibitor (LY294002, Calbiochem) or DMSO as a control. DNA lipofection or viral infection, followed by antibiotic selection for 1–3 weeks and FACS to obtain appropriate expression was used to generate stable lines of ectopic protein expression. Selective antibiotics were removed 24 h prior to 3D culture. For cells stably expressing EGFP-2xPH-TAPP1, 2 subclones with low cytoplasmic background were studied.

**Antibodies and immunocytochemistry**. Cysts immunolabelling was adapted from previously described protocols[12]: Cultures were fixed in 4% paraformaldehyde (PFA, Affimetrix) for 5–15 min at room temperature (RT), washed twice in PBS,

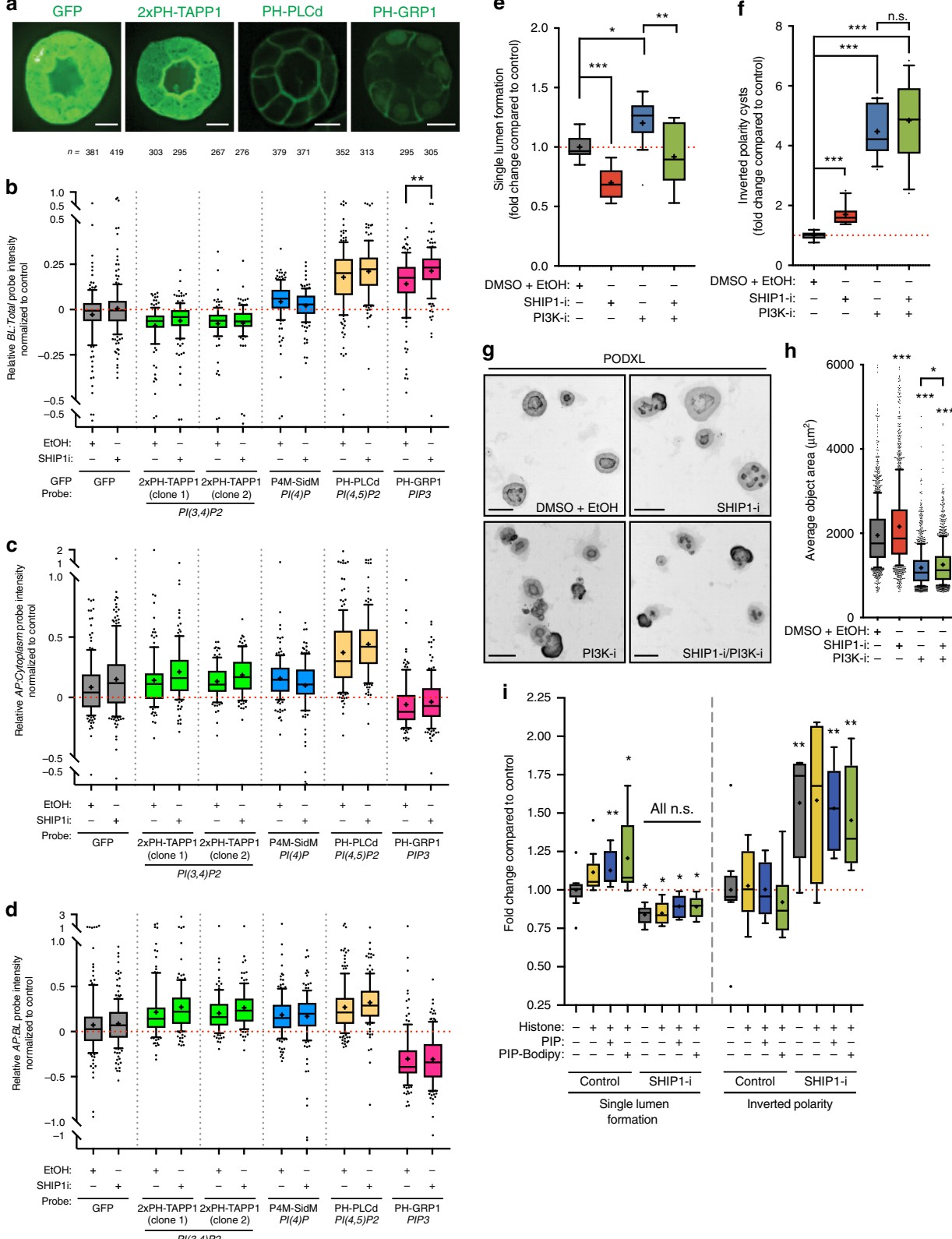

blocked for 1 h in PFS buffer (PBS, 0.7% w/v fish skin gelatin (Sigma-Aldrich), 0.5% saponin (Sigma-Aldrich)), and incubated with primary antibodies diluted in PFS at 4 °C overnight with gentle rocking. Then, cyst cultures were washed thrice with PFS and incubated with secondary antibodies diluted in PFS for 1 h at RT, followed by washing twice in PFS and twice in PBS. Primary antibodies are as described below. Alexa fluorophore-conjugated secondary antibodies (1:250) or Phalloidin (1:200) (both Invitrogen) and Hoescht to label nuclei (10 µg ml$^{-1}$), were

utilized. For validation of endogenous PI(3,4)P$_2$ staining in 3D, we optimized staining protocols. We tested four staining protocols in addition to the commercially recommended protocol (PI(3,4)P$_2$ Ab, Echelon, Z-P034), (i)[43], (ii)[12], (iii)[24], (iv)[44]. Buffer four produced endogenous labeling patterns mirroring EGFP-2xPH-TAPP1 and was used for all further PI(3,4)P$_2$ staining. Briefly, staining in buffer four is as follows: cysts were fixed in 4% PFA followed by three washes in glycine buffer (100 mM glycine in PBS), and two washes in PBS. Cysts were permeabilized

**Fig. 5** Regulation of PIP$_3$ basolateral levels is essential for de novo lumen formation. **a** Representative picture of cysts overexpressing specified PIP reporters. **b–d** Quantitation of relative basolateral to total **b**, apical to cytoplasm **c**, or apical to basolateral **d** PIP reporter intensity. All conditions are Log2 ratios normalized to GFP-overexpressing control MDCK cells, after treatment with either control (EtOH) or SHIP1 inhibitor. Box-and-whiskers: 10–90 percentile; +, mean; dots, outliers; midline, median; boundaries, quartiles. $n \geq 267$ cysts assessed from three replicate wells/condition/experiment, from three independent experiments. P-values: One-way ANOVA. **$P \leq 0.005$. **e**, **f** Quantitation of cysts displaying single lumen **e** or inverted polarity **f** after treatment with either control (DMSO + EtOH), SHIP1 inhibitor (SHIP1-i) and/or PI3K inhibitor (PI3K-i). Values are presented as a fold-change upon control. Box-and-whiskers: 10–90 percentile; +, mean; dots, outliers; midline, median; boundaries, quartiles. $n \geq 4000$ cysts assessed from 2 to 3 replicate wells/condition/experiment, from four independent experiments. P- values: Mann–Whitney test. *$P \leq 0.05$, **$P \leq 0.005$, ***$P \leq 0.0001$. **g** Representative images of cysts from **e** to **f** stained for Podxl (inverted greyscale). Scale bars, 50 μm. **h** Quantitation of average object area in treatments from **e** to **f**. $n \geq$ 4000 cysts, 2–3 replicate wells/condition/experiment, four independent experiments. Box-and-whiskers: 10–90 percentile; +, mean; dots, outliers; midline, median; boundaries, quartiles. P-values: One-way ANOVA. *$P \leq 0.05$, **$P \leq 0.005$, ***$P \leq 0.0001$. **i** Quantitation of either control or SHIP1-inhibited (SHIP1 inhibitor) single lumen (left) or inverted polarity (right) phenotypes, without or with histone carrier, with unlabeled or Bodipy-labeled PI (3,4)P$_2$. Values are fold change to control. Box-and-whiskers: 10–90 percentile; +, mean; dots, outliers; midline, median; boundaries, quartiles. $n \geq 2500$ cysts assessed from three replicate wells/condition/experiment, from three independent experiments. P-values: One-way ANOVA. *$P \leq 0.05$, **$P \leq 0.005$, ***$P \leq 0.0001$

in glycine buffer containing 0.1% of saponin for 20 min, and blocked in PBS containing 10% fetal calf serum and 0.1% saponin. Primary and secondary antibody incubation in blocking buffer were performed in the conditions specified above, followed by three washes in blocking buffer. List of used antibodies, in Supplementary Table 1.

**Image acquisition and analysis, PAPA and PAPI**. Confocal images were acquired either on Zeiss LSM 880 Airyscan confocal microscope, a Zeiss LSM 510 confocal, or an Opera Phenix Z9501 high-content imaging system (PerkinElmer). 3D culture has the imaging challenge of sparsely positioned objects (cysts) that can be positioned in different Z-planes. This has previously required manual imaging of each object, precluding large sample number analyses. We have overcome this by building two analysis pipelines to allow quantitation of cyst phenotypes or phosphoinositide distribution from hundreds to thousands of 3D cysts per condition: (1) a Pipeline for semi-Automated Polarity Analysis, PAPA, and (2) a Pipeline for semi-Automated Phosphoinositide Intensity analysis, PAPI. PAPA and PAPI make use of an Opera Phenix Z9501 high-content imaging system (PerkinElmer). To direct imaging only to cysts (and not the surrounding non-cyst areas) we used the PreciScan module (PerkinElmer) to scan entire wells to find cysts by performing z-stacks of each well at ×5 magnification, on-the-fly processing of images using user-defined rules to identify cysts, and then directing the microscope to perform high-resolution z-stacks of objects (cysts). In the case of PAPI, this requires cysts to express GFP-tagged PIP probes above a baseline level defined as optimum for imaging. At this step, we perform at least 8 optical sections every 2 μM, imaging at least 25 fields or objects (×20 and ×63, respectively). For PAPA, cysts are stained for apical marker (Podxl), the cortex (Phalloidin), whole-cell stain (cytoplasm), and nuclei (Hoescht). For PAPI, GFP-tagged PIP reporters were imaged in place of whole cell stain. For PAPA, as lumens can occur in different Z-planes of a given cyst, a maximum projection of the medial Z-planes was applied. Using Harmony imaging analysis software (PerkinElmer), user-defined phenotype classification rules to detect polarity orientation based on the localization and intensity of apical markers and the detection and quantification of number of lumens was applied to each cyst. For benchmarking of accuracy, PAPA was compared to manual imaging and counting of polarity phenotypes for control vs SHIP1-inhibited cysts. PAPA produced concordant quantitation of phenotypes to expert manual quantitation.

For PAPI, as maximum projection of different Z-planes could introduce quantification artefacts, analyses were performed only on the most medial Z-plane, which was automatically calculated based on the maximal luminal area. To detect PIP probe intensity in different subcellular locales, cysts were fixed and stained with Podxl to mark the apical domain, Phalloidin to mark the cortex, and Hoescht to mark the nucleus. Combinations of these stainings were used to create masked regions to calculate mean total, cortical, apical, basolateral, cytoplasmic (excludes nuclear region), and nuclear intensity per area in each region. For depiction of these regions, see Fig. 2a. Note that for all images taken at super-resolution for subcellular analysis, such as co-localization between Rab11 and PIPs or PIP-modifying enzymes, a Zeiss LSM 880 Airyscan confocal microscope was used from a single plane only. Statistical analysis was performed to calculate percentages of polarity phenotypes, normalized to control, and the relative PIP intensity in different cellular regions. Data were processed using KNIME analytics platform, and GraphPad Prism to generate graphs.

**Statistics**. Cyst phenotypes were binned into four categories: (a) single lumen, (b) multiple lumens/vesicular accumulation of Podxl, (c) inverted polarity, (d) other/no lumen. Relative percentages were normalized to control. For RNAi rescue experiments, only cysts expressing exogenous transgene were scored. Relative percentages from each category were normalized to control. Values are mean ± S.D. from 3 to 9 replicate experiments, with $n \geq 300$ cysts per replicate, unless otherwise

indicated. For qPCR, expression was normalized as fold change (log2) from the mean expression for all conditions. Significance was calculated using a paired, two-tailed Student's t-test, Mann–Whitney test, one-way ANOVA or two-way ANOVA test. No statistical methods were used to predetermine the sample size. No randomizations were used. The investigators were not blinded to allocation during experiments and outcome assessment. Statistical tests used are stated on every figure legend with P-values as appropriate. Data distribution should meet the normal distribution requirements. No estimate of variation. Data were analyzed using KNIME analytics platform, Excel (Microsoft) or Prism (Graphpad).

**Live cell dual-color imaging**. Live imaging was performed on cysts in eight-well chamber slides (Labtek II, Nunc). Cysts co-expressing EGFP-2xPH-TAPP1 and either TagRFP-T-Rab11a or Membrane-tdTomato were imaged on an inverted spinning-disk confocal microscope system (Yokogawa/Zeiss) with a 37 °C and 5% CO$_2$ controlled environment (Zeiss) and heated stage (PECON) through a 20 × 1.49NA lens (Zeiss), using 488/561 nm laser lines. Images were captured via an AxioCam Mrm (Zeiss). A stack of three images at 5 μm intervals was taken using an automated stage controlled via the ZEN software package (Zeiss). As cysts develop and move during image acquisition, the optimal focal plane was manually chosen for each time point post-acquisition from the stacked images, before compilation of image series of each movie. Movies and images were processed using ImageJ (NIH), as adapted from Stehbens et al.[45]. Briefly, the functions 'Remove Outliers' (radius, 2; threshold, 5), and 'Unsharp Mask' (radius, 7; mask, 0.5) were applied to improve image contrast.

**RNAi**. Lentiviral shRNAs were used to stably deplete proteins, described in Virus production and transduction methods section. ShRNA sequences were generated using iRNA software, as per Addgene (https://www.addgene.org/tools/protocols/plko/). RNAi target sequences are listed in Supplementary Table 2. Knockdown was verified by western blot or RT-qPCR procedures, normalized to GAPDH expression. Due to the canine origin of MDCK, shRNAs were custom-designed, and RNAi sequences chosen not to match human were utilized when rescuing with transgenes.

**RT-qPCR**. For detection of phosphoinositide kinases and phosphatases expression or for shRNA-mediated knockdown verification, RT-qPCR was used. Primers were designed based on the canine genome using Primer3Plus software and are listed in Supplementary Table 3. RT-qPCR was performed using EXPRESS One-Step SYBR® GreenER™ Universal (Thermo Fisher Scientific) per manufacturer's instructions.

**Virus production and transduction**. All lentiviruses were derived from the pLKO backbone. Plasmids were either co-transfected with Virapower packaging mix (Invitrogen) or co-transfected with pSPAX2 and pVSVG packaging plasmids, both using Lipofectamine 2000 into 293-FT cells according to manufacturer's instructions. Viral supernatants were collected 2 and 3 days after transfection, then clarified of cell debris using PES 0.45 μm syringe filters (Starlab) and concentrated using Lenti-X concentrator (Takara), following manufacturer's instructions. For retroviruses, viral supernatants were collected daily from days 5 to 7 after transfection of 293-GPG packaging cells. After centrifugation twice at 3500×g to remove cell debris, supernatants were snap-frozen in liquid nitrogen. For lentiviral transductions, MDCK cultures 18 h after plating were infected with viral supernatants for 12–16 h at 37 °C. Next, supernatants were diluted 1:1 with growth medium, cultured for a further 48 h at 37 °C, then passaged into appropriate antibiotic-containing medium. For retroviral transductions, cells 12–16 h post plating were incubated with viral supernatants supplemented with 10 μg ml$^{-1}$ Polybrene (Millipore) for 24 h at

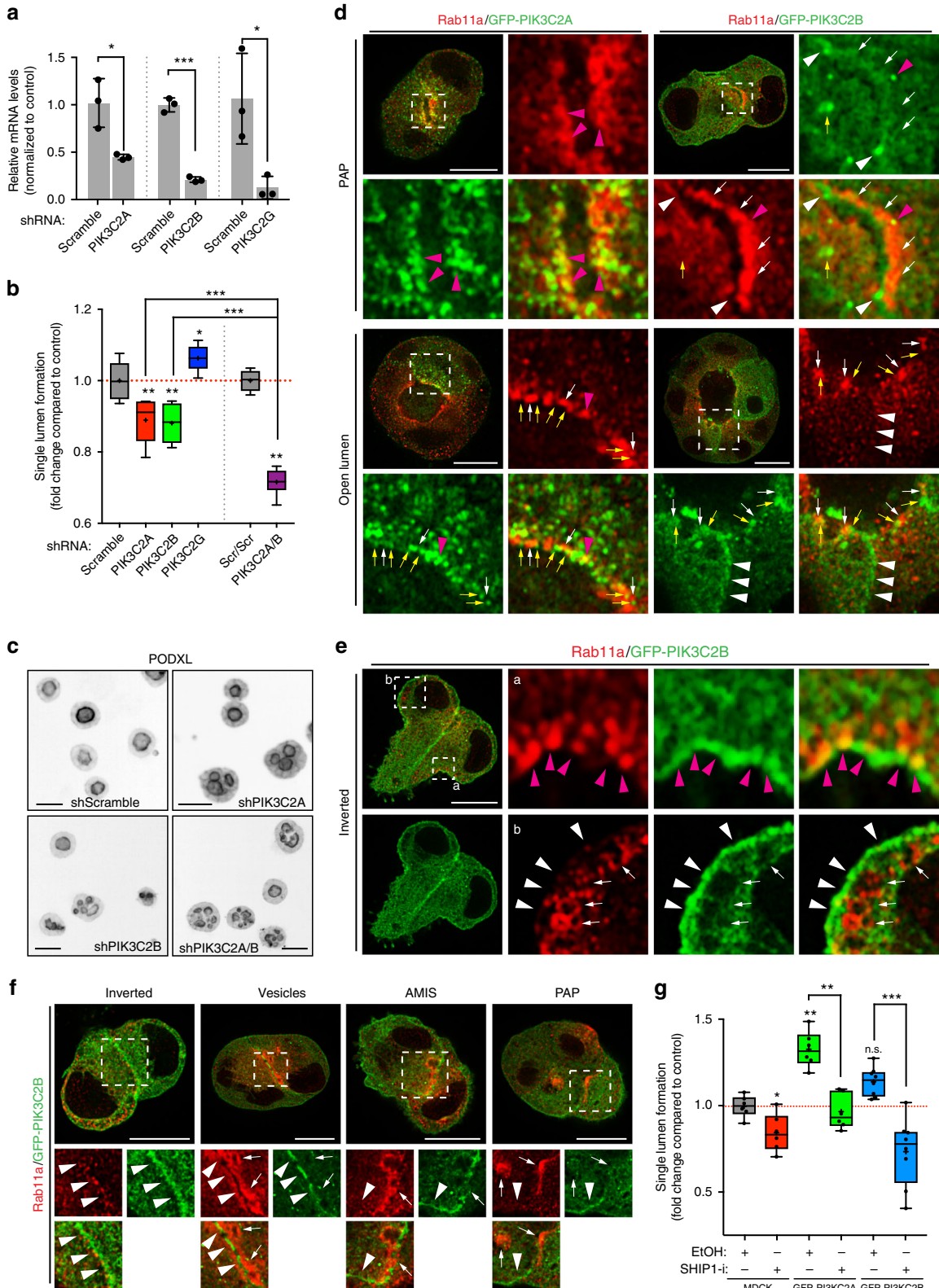

32 °C. Upon changing to fresh medium, cells were incubated for a further 48 h at 37 °C, before passage into appropriate antibiotic-containing medium. Hygromycin (0.5 mg ml$^{-1}$), blasticidin (12.5 μg ml$^{-1}$), puromycin (5 μg ml$^{-1}$), or zeocin (0.4 mg ml$^{-1}$) were used.

**Plasmids and cell lines**. The sources of stable cell lines or plasmids are: GFP-Rab11a[12]; GFP-PH-PLC[11]; GFP-PH-Grp1 (M. Birnbaum, University of

Pennsylvania, USA); GFP-PH-Akt[46]; GFP-2xFYVE(Hrs) (H. Stenmark, The Norwegian Radium Hospital, Norway); pEGFP-C1-P4M-SidM (Addgene Plasmid 51469), GFP-PH-FAPP1, GFP-OSH2-PH, GFP-OSH2-2xPH (T. Balla, NIH, USA); GFP-PH-ING2 (J. Yuan, Harvard Medical School, USA); GFP-PH-TAPP1 (D. Alessi, University of Dundee, UK); GFP-Cavin-1 (R. Parton, University of Queensland, Australia); GFP-Tubby-C (L. Shapiro, Columbia University, USA); GFP-PIK3C2A and GFP-PIK3C2B (V. Haucke, FMP, Germany); OCRL1-mCherry (Addgene plasmid 27675); SNX9-mCherry (Addgene

**Fig. 6** Class-II PI3-kinases regulate lumen formation. **a** Downregulation of *PIK3C2A*, *PIK3C2B* and *PIK3C2G* by shRNA. RNA extracts from MDCK cells stably expressing scramble or shRNA targeting the corresponding gene were analyzed by RT-qPCR. Mean ± s.d., $n = 3$ wells/condition, from one experiment. *P*-values: One-way ANOVA. *$P \leq 0.05$, ***$P \leq 0.0001$. **b**, **c** Quantitation of single lumen formation in cysts (48 h) stably expressing either scramble or PIK3C2A/B/G shRNAs alone or in combination **b** and representative images of Podxl staining from these condition (**c**, Podxl in inverted greyscale; scale bars, 50 μm). Values are fold change to control. Box-and-whiskers: 10–90 percentile; +, mean; dots, outliers; midline, median; boundaries, quartiles. $n \geq 1900$ cysts assessed from three replicate wells/condition/experiment, from two independent experiments. *P*-values: One-way ANOVA. *$P \leq 0.05$, **$P \leq 0.005$, ***$P \leq 0.0005$. **d–f** MDCK cysts stably expressing GFP-PIK3C2A or GFP-PIK3C2B (green), stained for Rab11a (red) at indicated polarity stages. Magenta arrowheads, PIK3C2A/B overlapping with Rab11a vesicles. Yellow arrows, Rab11a-negative, PIK3C2A/B punctate localization. White arrowheads, basolateral PIK3C2B. White arrows, Rab11 positive vesicles. Scale bars, 10 μm. **g** Quantitation of polarity phenotypes in control (EtOH) or SHIP1 inhibitor-treated (SHIP1-i) parental (MDCK) cysts, or stably expressing GFP-PIK3C2A or GFP-PIK3C2B. Box-and-whiskers: 10–90 percentile; +, mean; midline, median; boundaries, quartiles. PIK3C2A, $n \geq 1550$ cysts assessed from three replicate wells/condition/experiment, two independent experiments. *P*-values, Mann–Whitney test: *$P \leq 0.05$, **$P \leq 0.005$. PIK3C2B, $n \geq 800$ cysts assessed from 2 to 3 replicate wells/condition/experiment, three independent experiments

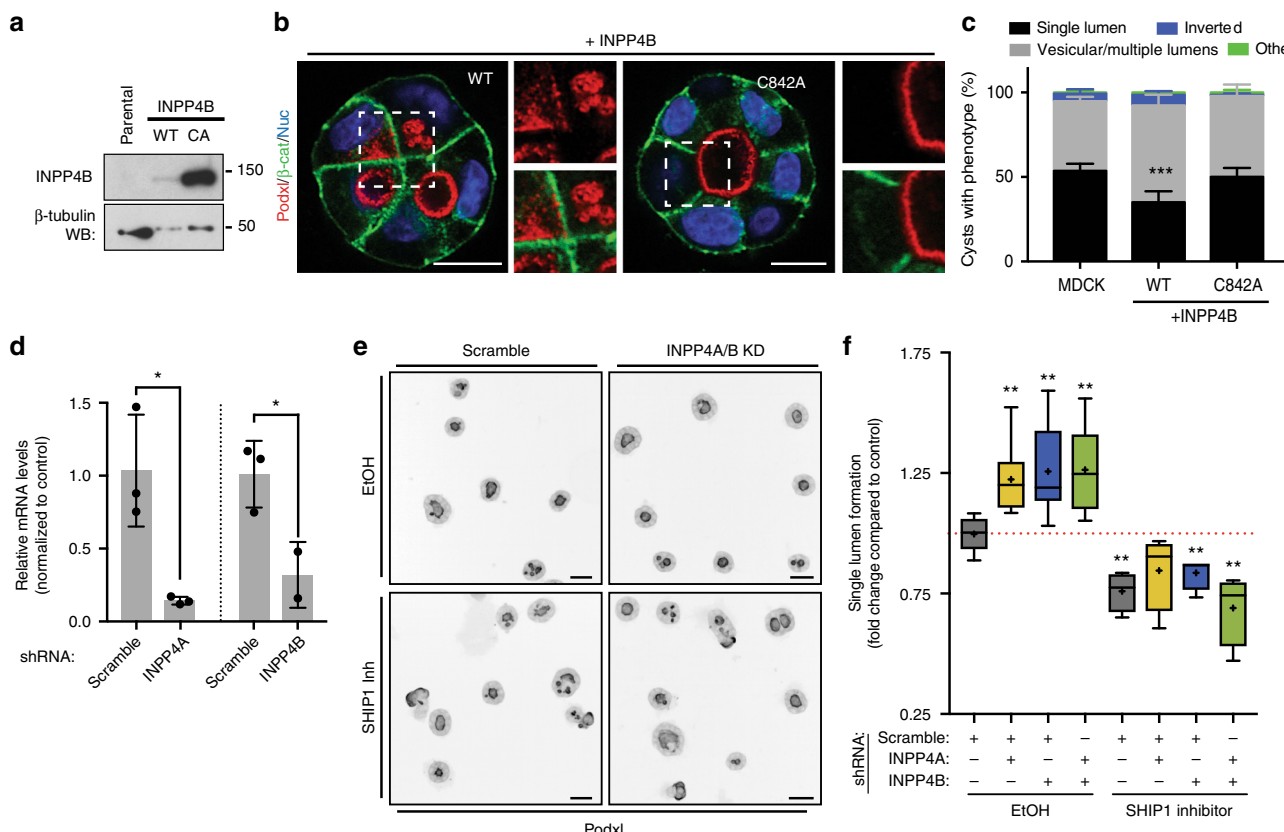

**Fig. 7** INPP4A/B negatively regulate apical polarization. **a** Analysis of parental MDCK or INPP4B WT or phosphatase-dead (CA, C842A) by western blot, showing INPP4B and β-tubulin expression. **b** Representative confocal images, cysts stained for Podxl (red), β-catenin (green) and nuclei (blue). mCherry (red). Note defective lumen formation, and subcortical Podxl accumulation upon INPP4B expression, but not the C842A mutant. Scale bars, 10 μm. **c** Phenotype quantitation without or with overexpression of WT or phosphatase-defective INPP4B. Mean ± s.d., $n \geq 300$ cysts assessed from three wells/ condition/experiment, three independent experiments. *P*-values: two-way ANOVA. ***$P \leq 0.0001$. **d** Downregulation of *INPP4A/B* by shRNA. RNA extracts from MDCK cells stably expressing scramble or shRNA targeting the corresponding gene were analyzed by RT-qPCR. Mean ± s.d., $n =$ three replicate wells/condition, from one experiment. *P*-values: One-way ANOVA. *$P \leq 0.05$. **e**, **f** Quantitation **f** of single lumen formation in 48 h control (EtOH) cysts stably expressing either scramble or INPP4A/B shRNAs, alone or in combination, and repeated in SHIP1-inhibited conditions (SHIP1-i) and representative images of Podxl staining from these condition (**e**, Podxl in inverted greyscale; scale bars, 50 μm). Values are fold change to control. Plots are box-and-whiskers: 10–90 percentile; +, mean; dots, outliers; midline, median; boundaries, quartiles. $n \geq 2000$ cysts assessed from three replicate wells/ condition/experiment, three independent experiments. *P*-values, Mann–Whitney test: *$P \leq 0.05$, **$P \leq 0.005$

plasmid 27678), EGFP-CDC42 WT, EGFP-CDC42Q61L, T23 PBD-YFP, GFP-Annexin2[11,12].

The following additional plasmids were generated through site-directed mutagenesis (Quikchange, Agilent) or standard subcloning: GFP-2xPH-TAPP1, eGFP-2xPH-TAPP1 R211L, EGFP-CDC42Q61L, TagRFPT-Rab11a (WT) in pQCXIH, mCherry-SNX9 KR (K267N, R327N), mCherry-SNX9 RYK (R286A, Y287A, K288A), mCherry-INPP4B (WT and C842A) in pmCherry-C1, SGFP2-

INPP5E in pSGFP2-C1 (Addgene Plasmid 22881) and in pRevTRE (Clontech), INPP5D-EGFP (WT and D673G) in pEGFP-N1. The complementary DNA templates for INPP5E, INPP4B and INPP5D were from Thermo Scientific (clone ID 5242186), Addgene plasmid 24324 and GE Healthcare (clone ID 8322634), respectively. For Membrane-tdTomato, the puromycin resistance cassette from pLKO-puro-Scramble, was replaced with Membrane-tdTomato (from Addgene plasmid 37351).

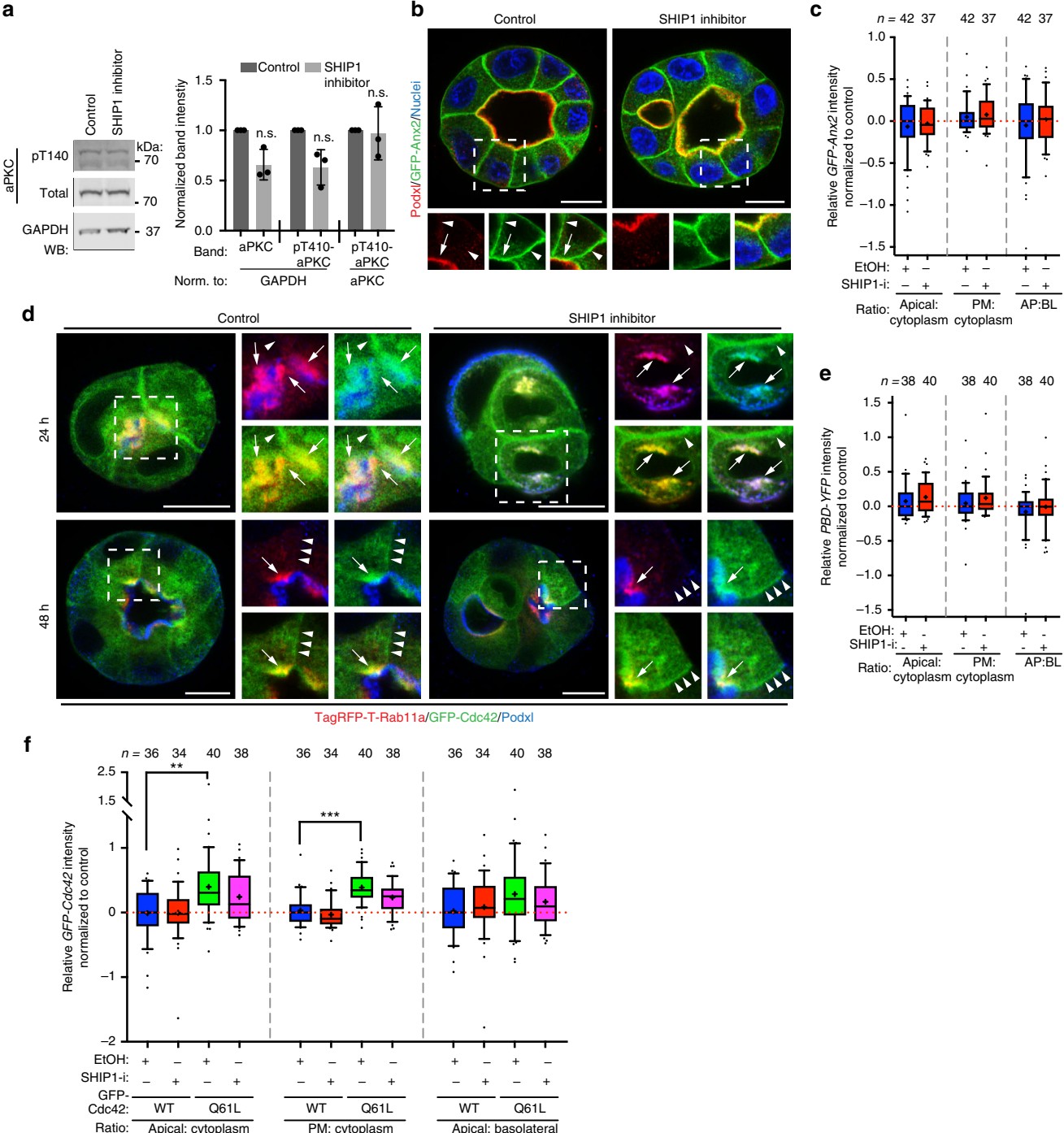

**Fig. 8** The apical Par complex is PI(3,4)P$_2$-independent. **a** Left, total cell lysates of 3D MDCK cysts upon treatment with SHIP1 inhibitor or ethanol as control, western blotted for total or phospho-aPKC (pT410), and GAPDH. Right, quantitation of normalized band intensity across experimental replicates. Mean ± s.d., $n = 3$, $P$-value, Mann–Whitney test. n.s. stands for non-significant. **b** Forty-eight hours MDCK cyst stably overexpressing GFP-Anx2 (green) treated with either ethanol (control) or SHIP1 inhibitor, stained for Podxl (red) and nuclei (blue). White arrows, apical membrane. White arrowheads, lateral domain. Scale bars, 10 μm. **c** Quantitation of apical to cytoplasm (left), plasma membrane to cytoplasm (center) or apical to basolateral (right) GFP-Anx2 relative intensity in control (Ethanol, EtOH) or SHIP1-inhibited (SHIP1-i) conditions. All conditions are Log2 ratios normalized to control cysts. Box-and-whiskers: 10–90 percentile; +, mean; dots, outliers; midline, median; boundaries, quartiles. $n \geq 37$ cysts per condition, assessed from one experiment. **d** Twenty-hour or 48 h MDCK cysts stably co-overexpressing GFP-Cdc42 (green) and TagRFPT-Rab11a (red) in control (EtOH) or SHIP1-inhibited conditions, stained for Podxl (blue). White arrows, Rab11a/Cdc42 co-localization. White arrowheads, basolateral. Scale bars, 10 μm. **e**, **f** Quantitation of relative apical to cytoplasm (left), plasma membrane to cytoplasm (center), or apical to basolateral (right) intensity of MDCK cysts stably expressing either PBD-YFP **e** or GFP-Cdc42 (WT or Q61L mutant, **f**) in control (EtOH) or SHIP1-inhibited (SHIP1-i) conditions. All conditions are Log2 ratios normalized to control cysts. Box-and-whiskers: 10–90 percentile; +, mean; dots, outliers; midline, median; boundaries, quartiles. $n \geq 34$ cysts per condition, assessed from one experiment. $P$-value, Mann–Whitney test. **$P \leq 0.005$, ***$P \leq 0.0001$

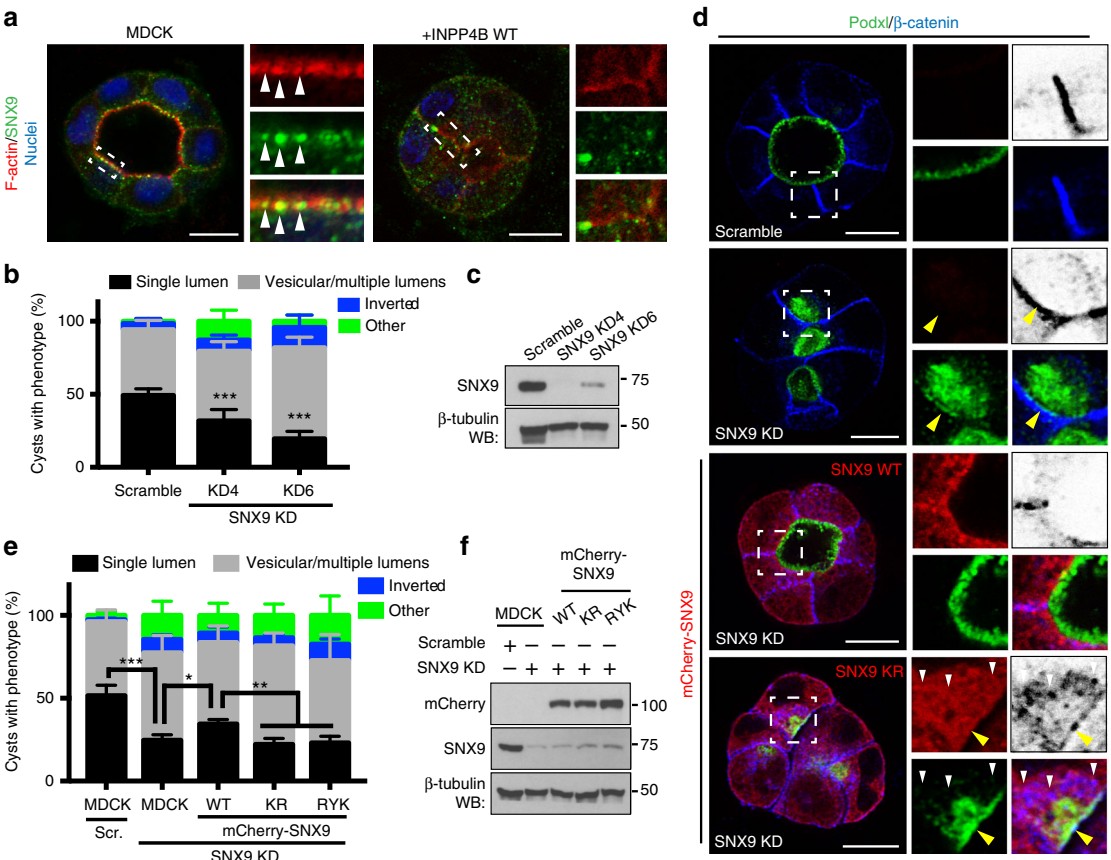

**Fig. 9** SNX9-PI(3,4)P$_2$ association is essential for lumen formation. **a** SNX9, F-actin and nuclei labeling in cysts at 48 h without or with INPP4B WT overexpression. White arrowheads, subapical SNX9 puncta. **b** Quantitation of cyst phenotypes upon control (scramble) or SNX9 KD using two independent shRNAs. Mean ± s.d., $n \geq 300$ cysts assessed from three wells/condition/experiment, three independent experiments. P-values: two-way ANOVA. ***$P \leq 0.0001$. **c** Western blot of SNX9 and β-tubulin expression in cells stably expressing either scramble or SNX9 shRNAs. **d** Control (scramble) and SNX9-depleted cysts without or with co-expression of RNAi-resistant WT or PIP-binding defective mutant (KR) mCherry-SNX9 (red) stained for Podxl (green) and β-catenin (blue, inverted greyscale in cropped regions). Yellow arrowheads, basolateral; white arrowheads, vesicular β-catenin and SNX9. **e** Quantitation of cysts phenotypes in control (scramble) or SNX9-depleted cysts without or with expression of RNAi-resistant wild-type mCherry-SNX9 or PIP-binding defective mutants of SNX9 (KR and RYK). Mean ± s.d., $n \geq 300$ cysts assessed from three wells/condition/experiment, three independent experiments. P-values: two-way ANOVA. *$P \leq 0.05$, **$P \leq 0.005$, ***$P \leq 0.0001$. **f** Total cell lysates of conditions described in **e** western blotted for mCherry, SNX9, and β-tubulin. All scale bars, 10 μm

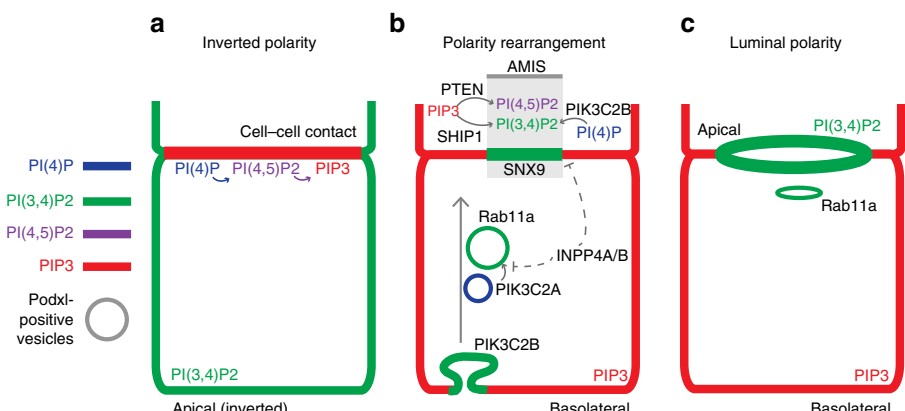

**Fig. 10** Model of PIP rearrangements during lumen formation. Shortly after plating into 3D, single MDCK cells undergo morphogenesis to form a cell doublet with initially inverted polarity **a**, whereby the apical surface marked by Podxl abuts the ECM. At this stage, a PI(4)P>PI(4,5)P$_2$ > PIP$_3$ cascade defines the cell–cell contact, while PI(3,4)P$_2$ is inverted. **b** To reorient polarity, peripheral Podxl internalizes into vesicles which are initially PI(3)P and/or PI(4)P positive, before progressively enriching to become PI(3,4)P$_2$/Rab11a-positive. These vesicles target to the apical membrane initiation site (AMIS). The PI(3,4)P$_2$-promoting Class-II PI3Ks PIK3C2A/B promote, whereas the PI(3,4)P$_2$-degrading phosphatases INPP4A/B inhibit, this process. Concomitantly, the AMIS is formed by two PIP phosphatases, PTEN and SHIP1, which focally convert PIP$_3$ into, PI(4,5)P$_2$ or PI(3,4)P$_2$, respectively to form this zone for apical vesicle delivery. **c** Upon delivery of transcytosing Podxl, de novo lumen formation is completed, leaving cortical PI(4)P and PI(4,5)P$_2$, basolateral PIP$_3$, and apical and recycling endosome PI(3,4)P$_2$

**Immunoblotting**. Protein blotting was adapted from previous protocols[13]. Ice-cold extraction buffer (50 mM Tris-HCl, pH 7.4, 150 mM NaCl, 0.5 mM $MgCl_2$, 0.2 mM EGTA, and 1% Triton X-100 plus 50 mM NaF, 1 mM $Na_3VO_4$ and complete protease inhibitor cocktail tablet (Roche, Mannheim, Germany)) on ice for 5 min was used for cell solubilization. Cells were then scrapped and passed through a 27½ -gauge needle before extraction at 4 °C for 25 min. Clarification of post-nuclear supernatants was performed by centrifugation at $14,000 \times g$ at 4 °C for 10 min. SDS-PAGE was used to separate samples, followed by transfer to PVDF membranes. Western analysis was performed using either chemiluminescence (Super-Signal Chemiluminescence Kit; Pierce, Rockford, IL) or infrared fluorescent secondary antibodies and quantitative detection (Odyssey CLx, Li-COR Biosciences). A BCA Protein Assay Reagent kit (Pierce) was used to determine protein concentration. Transfer and protein loading were monitored by staining 0.1% Coomassie Brilliant Blue. Statistical analysis of western blots and generation of accompanying graphs were performed using Excel (Microsoft) or Prism (Graphpad). Uncropped membranes are presented in Supplementary Figure 5.

**Delivery of exogenous PI(3,4)P₂**. Exogenous PIP was delivered to cysts as previously reported[11] with appropriate adaptations: $PI(3,4)P_2$ and Histone carriers (Shuttle $PIP^{TM}$, Echelon, Salt Lake City, UT) were freshly prepared in PIP solution (150 mM NaCl, 4 mM KCl, 20 mM HEPES at pH 7.2) to a final concentration of 300 or 100 μM, respectively. Complexes were combined for 10 min, and diluted 1:10 in Hank's buffered salt solution, solution added for 30 min to MDCK cysts that had previously been cultured in 3D for 48 h. Cells were fixed, stained, and imaged.

**Repeatability of experiments**. Immunofluorescence images of 3D MDCK cysts: representative image from one field of $\sim 1 \times 10^4$ cells in shown. Each experiment was repeated at least twice unless otherwise indicated.

Immunofluorescence images of MDCK cells grown in Transwell: representative image from two image sets.

Movies: Representative movie from five movies of single lumen MDCK cysts.

Western blot:

Figure 4f: representative image from two blots.

Figure 7a: blot, performed one time

Figures 5c, f: representative image from three blots.

Figure 8a: representative image from three blots.

Supplementary Figure 1b: representative image from two blots.

Polarity analysis (Manual or PAPA):

Figure 4d: Left, three wells/condition/experiment, four independent experiments, manual analysis. Right, three wells/condition, three independent experiments, PAPA analysis.

Figure 4g: Four wells/condition/experiment, manual analysis.

Figure 5e, f, h: 2–3 wells/condition/experiment, four independent experiments, PAPA analysis.

Figure 5i: Three wells/condition/experiment, three independent experiments, PAPA analysis.

Figure 6b: Three wells/condition/experiment, two independent experiments, PAPA analysis.

Figure 6g: 2–3 wells/condition/experiment, 2 or 3 independent experiments, PAPA analysis.

Figure 7c: Three wells/condition/experiment, three independent experiments, manual analysis.

Figure 7f: three wells/condition/experiment, three independent experiments, PAPA analysis.

Supplementary Figure 1c. Three wells/condition/experiment, one experiment, left manual, right PAPA analysis.

Supplementary Figure 3h: Three wells/condition/experiment, five independent experiments, manual analysis.

PAPI analysis:

Figure 2c: Three wells/condition/experiment, 2–3 independent experiments

Figure 5b–d: Three wells/condition/experiment, Three independent experiments

Figure 8c, e, f: ≥25 independent objects/condition, one independent experiment.

## Data availability
All relevant data are available from the authors upon request.

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

## Acknowledgements

We thank the many investigators that shared reagents. Supported by NIH grants R01DK074398 and R01DK091530 and Baltimore Polycystic Kidney Disease Research and Clinical Core Center P30 DK090868 (K.M.), CRUK C596/A17196 (E.S.), and NIH K99CA163535 (D.B.). We thank C. Winchester for helpful discussions.

## Author contributions

A.R.-F., J.R. and D.M.B. designed experiments and analyzed data. D.M.B. wrote the manuscript. A.R.-F., J.R., E. Sandilands, M.N., M.A.M. and D.M.B. performed experiments. L.M. and E. Shanks helped with development of high-throughput imaging and analysis. K.E.M. and D.M.B. supervised the study. All authors discussed the results and implications and commented on the manuscript at all stages.

## Additional information

**Competing interests:** The authors declare no competing interests.

