## [Transparent Peer Review File · Nature Communications]

Editorial Note: This manuscript has been previously reviewed at another journal that is not operating a transparent peer review scheme. This document only contains reviewer comments and rebuttal letters for versions considered at Nature Communications. Mentions of prior referee reports have been redacted.

Reviewers' comments:

Reviewer #1 (Remarks to the Author):

The authors present an extensively revised version of their earlier Ms, in which they have carefully addressed all my previous questions and concerns. I thus enthusiastically endorse publication of this interesting, yet complex, story in Nature Communications.

Reviewer #2 (Remarks to the Author):

I reviewed this paper previously [redacted]. At that time I raised a series of points that the authors have successfully responded to. The new PIPA technology introduced in this manuscript is an excellent addition to the field.

Reviewer #3 (Remarks to the Author):

The authors have worked very hard to improve this manuscript since I first reviewed it. There have been clear efforts to improve the quality of the images, use of independent reporters of PI(3,4)P2, the clarity of the description of the imaging technologies, improved quantitation of the images and new experiments using exogenous PI(3,4)P2 to test its roles in lumen formation/polarisation.

Nevertheless, I still have several issues with the current manuscript.

A recent publication indicates that PTEN can act as a PI(3,4)P2 3-phosphatase (Malek M. et al Mol Cell 2017). Although difficult to reconcile with the model proposed in this study, the conclusions of Malek et al do not invalidate the model / work in this paper, because, for example in some contexts PTEN may only display PIP3 or PI(3,4)P2 phosphatase activity etc. However, it should at least be discussed.

Therefore PI(3,4)P2 would be expected to be low where PTEN accumulates; eg in the AMIS -as defined by the paper cited by this study (Chou S et al 2016 Nature Comm); and yet the here the authors are apparently reporting accumulation of PI(3,4)P2 in the AMIS.

The role of PTEN in polarisation/lumen formation in these types of model is very difficult to untangle. Loss of PTEN across a number of 3D models (eg MCF10a cells forming acini in Matrigel, eg Gewinner C. et al Cancer Cell 2009) leads to the formation of solid-cell-cell-filled acini rather than cavity-filled acini. This is probably a result of a combination of cell "over-growth" and reduced apoptosis. These issues combine with any roles for PTEN in polarisation/lumen formation to create complex changes in cyst structure that can be read-out in a variety of ways, like those used in this paper. This issue arises with any manipulation, not just loss of PTEN, that might impact cell growth and survival. To interpret all of the changes in polarisation/lumen formation simply in terms of impacts on that pathway is too simplistic. I acknowledge I cannot see any easy solution to this problem other than testing the impact of manipulations that are known to cause changes in cell growth / survival but no changes in polarisation/lumen formation or vice versa.

The authors argue that their intensive image analysis pipeline validates the signal detected by the

TAPP-1 reporter because of the significant difference between images decorated with GFP compared to GFP-wild-type-TAPP and lack of significant difference between GFP and a lipid-non-binding mutant of GFP-TAPP. The key statistical comparison is between the mutant and the wild-type reporters and that is not provided but does not look significant. This issue is confounded by the striking lack of nuclear staining by the mutant TAPP construct compared to GFP and the wild-type construct and the potential for that difference to contribute to broader differences measured by the image analysis between GFP and wild-type TAPP. The authors now use an anti-PI(3,4)P2 antibody to support their use of the TAPP reporter. This is an important additional independent control, but in the absence of appropriate internal controls demonstrating that their staining with the antibody is specific (eg lost with inhibition of PI(3,4)P2 synthesis or increased by methods that cause PI(3,4)P2 accumulation) that evidence alone does not nullify any concerns about interpretation of results obtained with the TAPP-reporter.

The improved images now make it possible to see some of the claimed co-localisations reported in this paper are more likely to represent coincidence based on "vertical overlay" in the optical sections. This is mostly the case for the "punctate" distributions; eg Fig5D (overlap of Rab11a and PIK3C2). It does not appear this is a result of a machine-fault leading to misalignment of images captured at different wavelengths. To be clear, I am NOT expecting a substantial proportion of the reporters to co-localise.

We thank all reviewers for their enthusiastic support of our manuscript. We provide point-by-point response to reviewer 3's comments, which we agree are important to clarify. We have indicated any changes in the manuscript and supplementary information file by colouring text in red.

Reviewer #3.

The authors have worked very hard to improve this manuscript since I first reviewed it. There have been clear efforts to improve the quality of the images, use of independent reporters of PI(3,4)P₂, the clarity of the description of the imaging technologies, improved quantitation of the images and new experiments using exogenous PI(3,4)P₂ to test its roles in lumen formation/polarisation.

We thank the reviewer for their positive comments and recognising the amount of work that we put into providing a revised manuscript.

Nevertheless, I still have several issues with the current manuscript.

Comments:

1a. A recent publication indicates that PTEN can act as a PI(3,4)P₂ 3-phosphatase (Malek M. et al Mol Cell 2017). Although difficult to reconcile with the model proposed in this study, the conclusions of Malek et al do not invalidate the model / work in this paper, because, for example in some contexts PTEN may only display PIP₃ or PI(3,4)P₂ phosphatase activity etc. However, it should at least be discussed.

Therefore PI(3,4)P₂ would be expected to be low where PTEN accumulates; eg in the AMIS -as defined by the paper cited by this study (Chou S et al 2016 Nature Comm); and yet the here the authors are apparently reporting accumulation of PI(3,4)P₂ in the AMIS.

We agree that this important paper and point should be discussed. We now include the following paragraph in the discussion (in red text):

“PTEN can also function as a PI(3,4)P₂ phosphatase. Given that PTEN is enriched at the forming AMIS, whether it is acting solely on PIP₃ or also on PI(3,4)P₂ is unknown. If both are occurring, the AMIS localization of PIK3C2B may help to focally reverse this dual specificity, leading to co-enrichment of apical PI(4,5)P₂ and PI(3,4)P₂. The exact function of PTEN at the AMIS thus warrants further attention.”

1b. The role of PTEN in polarisation/lumen formation in these types of model is very difficult to untangle. Loss of PTEN across a number of 3D models (eg MCF10a cells forming acini in Matrigel, eg Gewinner C. et al Cancer Cell 2009) leads to the formation of solid-cell-cell-filled acini rather than cavity-filled acini. This is probably a result of a combination of cell “over-growth” and reduced apoptosis. These issues combine with any roles for PTEN in polarisation/lumen formation to create complex changes in cyst structure that can be read-out in a variety of ways, like those used in this paper. This issue arises with any manipulation, not just loss of PTEN, that might impact cell growth and survival. To interpret all of the changes in polarisation/lumen formation simply in terms of impacts on

that pathway is too simplistic. I acknowledge I cannot see any easy solution to this problem other than testing the impact of manipulations that are known to cause changes in cell growth / survival but no changes in polarisation/lumen formation or vice versa.

We agree that this is an important point – *the extent to which polarity and growth regulated by PIPs can be molecularly uncoupled* - which we and others in the field try to grapple with. We likewise do not have an easy solution.

Although we do see modest changes in cyst area when we modify PIP enzymes (for example, see Fig 4H), we do not see overgrowth as an obligate phenotype when we increase PI(3,4)P2 or PIP3 levels. Thus, we argue that the focus on polarity phenotypes is not too simplistic, but rather is the main phenotype in this system.

The main point here may be ‘in this system’. Our experience is that context matters; in the MCF-10A system referenced by the reviewer, these already form lumens poorly (in fact, they lack tight junctions and the Crumbs3 polarity complex). Thus, they are primed to be affected by pathways that perturb luminal apoptosis, such as PTEN depletion. In our unpublished work, genetically PTEN-null prostate primary mouse organoids have no apparent overgrowth phenotype. Therefore, our current understanding is that it is the context of the system in which PIPs are altered that determines whether the primary defect is *i)* polarity or *ii)* polarity and growth.

We believe that this point is something far beyond a single manuscript and certainly not the main focus of this manuscript. However, to at least mention this consideration we now include the following paragraph in the **Discussion** (red text):

“In addition to their function in cell polarization, phosphoinositides participate in cell growth and survival pathways. For instance, correct cell polarization may lay upstream to decisions of cell death or survival during morphogenesis. Likewise, perturbation of PIP-modifying enzymes, such as the PTEN or INPP4B loss observed in cancer, may be facilitative of the disrupted polarity and overgrowth of tumours. The extent to which these processes are distinct pathways or are intimately linked remains unclear.

2. The authors argue that their intensive image analysis pipeline validates the signal detected by the TAPP-1 reporter because of the significant difference between images decorated with GFP compared to GFP-wild-type-TAPP and lack of significant difference between GFP and a lipid-non-binding mutant of GFP-TAPP. The key statistical comparison is between the mutant and the wild-type reporters and that is not provided but does not look significant.

We apologise that this was not included previously, due to a desire for simplifying complex graphs with multiple comparisons. We update **Fig S2C** with the requested analysis.

For **Basolateral-to-Total**, and **Nuclear-to-Total** ratios the PIP-binding-deficient 2xPH-TAPP1 mutant is significantly altered compared to either GFP, or to both clones of the WT probe. For **Apical-to-Cytoplasm** ratio the mutant probe is significantly different compared to clone 2. The mutant versus clone 1 sits on the fence of significance. Using the rigorous statistical

methods we apply to PAPI analysis data (One-way ANOVA with correction for multiple testing), this difference is not significant only for this combination. However, if using an alternate **and equally appropriate** statistical testing methods (such as an unpaired t-test with Welch's correction), this is indeed significant. We prefer to leave this as not significant as defined by the more arduous ANOVA method. Note that *BOTH* WT probe clones show significant **AP: Cytoplasm** enrichment vs GFP alone, whereas the mutant *DOES NOT*.

Part of the reason we made a second clone of the TAPP1 probe is that clone 1 showed a mixed population of cysts with varied expression of the probe, whereas clone 2 showed robust and uniform apical enrichment. We believe that this reason underlies the statistical discrepancies described above. For the sake of good science practice and rigorous analysis we included such data from both clones, as we believe that the cumulative data support our conclusions for apical enrichment of PI(3,4)P2. For the sake of clarity, we can simply remove clone 1 if it is more confusing than helpful.

This issue is confounded by the striking lack of nuclear staining by the mutant TAPP construct compared to GFP and the wild-type construct and the potential for that difference to contribute to broader differences measured by the image analysis between GFP and wild-type TAPP.

The following points refer to Fig. S2C.

We had taken this point already into consideration in our PAPI analyses. Our measure of **Apical: Cytoplasm** ratio specifically excludes the nuclear labelling, thus negating such a concern for this ratio. The **cytoplasm** mask is defined as the cellular region remaining after the plasma membrane, lumen, and nuclei regions were subtracted. Thus, the differences in nuclear labelling with the WT vs mutant PIP probe do not contribute to this ratio.

For **Nuclear: Total** ratio, we need to include the nucleus, so it is appropriate to use **Total** intensity rather than the cytoplasm mask. This ratio shows a significant and robust *reduction* of nuclear intensity for the mutant when compared to GFP or either WT clone.

For **Basolateral: Total** ratio, the decrease in basolateral intensity of both WT clones compared to control (GFP) is maintained if we instead compare the **Basolateral: Cytoplasm** ratio (not shown). What does change is the directionality, but not significance, of only the mutant probe. The **Basolateral: Cytoplasm** ratio for the mutant instead shows significantly less basolateral intensity than GFP, but no significant difference to the WT (both clones). Thus, the reviewer is correct that in this instance the method of intensity ratio calculation changes the interpretation of whether the mutant but not the WT probe becomes basolaterally enriched. Given that the main question is whether the probe is apical, rather than *the degree to which it is less basolateral than GFP alone*, we can update this point if deemed necessary with whichever interpretation is requested by the reviewer.

The regions being compared in each of these ratios are depicted in **Fig. S2A, section iv**. We apologise as we see that this point was not so clear in the previous manuscript. For clarification, we update the following section:

In Extended Experimental Procedures, Image acquisition and analysis, PAPA and PAPI section.

“Combinations of these stainings were used to create masked regions to calculate mean total, cortical, apical, basolateral, cytoplasmic (excludes nuclear region), and nuclear intensity per area in each region. For depiction of these regions, see Fig. S2A.”

The authors now use an anti-PI(3,4)P2 antibody to support their use of the TAPP reporter. This is an important additional independent control, but in the absence of appropriate internal controls demonstrating that their staining with the antibody is specific (eg lost with inhibition of PI(3,4)P2 synthesis or increased by methods that cause PI(3,4)P2 accumulation) that evidence alone does not nullify any concerns about interpretation of results obtained with the TAPP-reporter.

We have gone to great lengths to validate *in our system* apical PI(3,4)P2 localisation.

The reviewer requests validation of the specificity of the anti-PI(3,4)P2 antibody utilised. We point out that this antibody is extensively validated¹⁻⁵, including that it shows the same localisation as the most widely used PI(3,4)P2 probe, 2xPH-TAPP1. Given the extensive use of both this probe and this antibody in the literature, our demonstration that the GFP-2xPH-TAPP1 probe is (i) apically enriched through quantitative analysis (Fig S2B-C), (ii) is dependent on phosphoinositide binding (Fig S2B-C), and (iii) overlaps with endogenous PI(3,4)P2 immunolabelling (Fig S2 D-H), we believe that we have already provided extensive evidence to support our assertions.

It would simplify our manuscript to have a scenario where we decrease production of PI(3,4)P2, and show less apical probe or antibody labelling. The problem is that by blocking PI(3,4)P2 production we also block formation of the very domain at which we need to measure (*i.e.* the apical lumen). One cannot quantify the levels of a PIP in the apical surface if there is no apical surface.

One can find individual cysts that have increased or decreased apical PI(3,4)P2 levels upon PIP-modifying enzyme manipulation (for examples see Fig. 4, outliers on the graphs). Rather than provide such anecdotal example images, we provide in-depth, robust statistical analysis of whether this holds true over hundreds of cysts. Rather than forcing a hypothesis onto the data (*i.e.* that we can increase/decrease levels of a PIP and the domain at which we need to measure will be unchanged), we believe that the data should be able to speak for itself. Changing PI(3,4)P2-producing enzymes alters the formation of PI(3,4)P2-enriched domains. But, if an apical domain forms, it contains a defined level of PI(3,4)P2 (Fig 4, S2). Therefore, the requested validation experiments are not possible in a 3D system where the alterations performed to ‘test specificity’ do not have a benign effect on morphogenesis.

Whilst we understand that this may still not nullify all concerns of the reviewer, we believe that the extensive data we provide, coupled to the common use of this probe and antibody in the literature, cumulatively support our conclusions.

3. The improved images now make it possible to see some of the claimed co-localisations reported in this paper are more likely to represent coincidence based on “vertical overlay” in the optical sections. This is mostly the case for the “punctate” distributions; eg Fig5D (overlap of Rab11a and PIK3C2). It does not appear this is a result of a machine-fault leading to misalignment of images captured at different wavelengths. To be clear, I am NOT expecting a substantial proportion of the reporters to co-localise.

For our co-localisations to represent vertical overlay in optical sections, different channels would have to be obtained in consecutive Z-planes, and projected into a single pseudo-plane. This is not the case. Images for co-localisation from Fig 5D (as well as Fig 2A, 2C, 5F, S2B S2D-H, 3C, S6B, S6D) were all obtained using a Zeiss 880 Airyscan super-resolution confocal in the same Z-plane. Thus, such artefactual colocalization is extremely unlikely. Instead, our co-localisations are *bona fide* colocated labelling/fluorescence from these proteins.

In assessing why the reviewer might assert such an unlikely scenario, we assume that our description of imaging methodologies had not made this clear. We apologise if our description of using multiple Z-planes for projecting cyst phenotypes for machine learning made it seem as if we used multiple Z-planes for intracellular co-localisation. This is not the case. To clarify this, we have amended our description of imaging methods.

In Extended Experimental Procedures, Image acquisition and analysis, PAPA and PAPI section, we include the following sentence:

‘Note that for all images taken at super-resolution for subcellular analysis, such as co-localisation between Rab11 and PIPs or PIP-modifying enzymes, a Zeiss LSM 880 Airyscan confocal microscope was used from a single plane only.’

Additional notes to reviewer and editor:

Whilst this review was occurring we performed additional replicates of some experiments. Figures 3D and 5G have been updated with these extra data points. Note that these do not change the conclusion of the data, but merely increase the sample size.

References:

1. Posor, Y. *et al.* Spatiotemporal control of endocytosis by phosphatidylinositol-3,4-bisphosphate. *Nature* **499**, 233-237 (2013).
2. Malek, M. *et al.* PTEN Regulates PI(3,4)P2 Signaling Downstream of Class I PI3K. *Molecular cell* **68**, 566-580 e510 (2017).
3. Braccini, L. *et al.* PI3K-C2gamma is a Rab5 effector selectively controlling endosomal Akt2 activation downstream of insulin signalling. *Nat Commun* **6**, 7400 (2015).
4. Marat, A.L. *et al.* mTORC1 activity repression by late endosomal phosphatidylinositol 3,4-bisphosphate. *Science* **356**, 968-972 (2017).
5. Boucrot, E. *et al.* Endophilin marks and controls a clathrin-independent endocytic pathway. *Nature* **517**, 460-465 (2015).

REVIEWERS' COMMENTS:

Reviewer #3 (Remarks to the Author):

The manuscript has been further improved. The more complete discussion of some of the uncertainty involved in the work and model make for a more accurate and balanced paper. Additional clarifications of statistics and methods also helped.